# Wasserstein Transfer Learning

**Kaicheng Zhang**[1]* **Sinian Zhang**[2]* **Doudou Zhou**[3]† **Yidong Zhou**[4]†

[1]School of Mathematical Sciences, Zhejiang University, China
[2]Division of Biostatistics and Health Data Science, University of Minnesota, USA
[3]Department of Statistics and Data Science, National University of Singapore, Singapore
[4]Department of Statistics, University of California, Davis, USA
3210102033@zju.edu.cn, zhan9381@umn.edu,
ddzhou@nus.edu.sg, ydzhou@ucdavis.edu

## Abstract

Transfer learning is a powerful paradigm for leveraging knowledge from source domains to enhance learning in a target domain. However, traditional transfer learning approaches often focus on scalar or multivariate data within Euclidean spaces, limiting their applicability to complex data structures such as probability distributions. To address this limitation, we introduce a novel transfer learning framework for regression models whose outputs are probability distributions residing in the Wasserstein space. When the informative subset of transferable source domains is known, we propose an estimator with provable asymptotic convergence rates, quantifying the impact of domain similarity on transfer efficiency. For cases where the informative subset is unknown, we develop a data-driven transfer learning procedure designed to mitigate negative transfer. The proposed methods are supported by rigorous theoretical analysis and are validated through extensive simulations and real-world applications. The code is available at https://github.com/h7nian/WaTL.

## 1 Introduction

In recent years, transfer learning [32] has emerged as a powerful paradigm in machine learning, enabling models to leverage knowledge acquired from one domain and apply it to related tasks in another. This approach has proven especially valuable in scenarios where data collection and labeling can be costly, or where tasks exhibit inherent similarities in structure or representation. While early successes focused on conventional data types such as images [30], text [28], and tabular data [38], there is growing interest in extending these methods to more complex data structures. Such data often reside in non-Euclidean spaces and lack basic algebraic operations like addition, subtraction, or scalar multiplication, posing challenges for traditional learning algorithms. A key example is probability distributions [27], where for example the sum of two density functions does not yield a valid density.

Samples of univariate probability distributions are increasingly encountered across various research domains, such as mortality analysis [14], temperature studies [44], and physical activity monitoring [23], among others [27]. Recently, there has been a growing focus on directly modeling distributions as elements of the Wasserstein space, a geodesic metric space related to optimal transport [36, 24]. The absence of a linear structure in this space motivates the development of specialized transfer learning techniques that respect its intrinsic geometry.

To address this gap, we introduce *Wasserstein Transfer Learning* (WaTL), a novel transfer learning framework for regression models where outputs are univariate probability distributions. WaTL

---

*Equal contribution.
†Corresponding authors. Doudou Zhou and Yidong Zhou contributed equally.

39th Conference on Neural Information Processing Systems (NeurIPS 2025).

effectively leverages knowledge from source domains to improve learning in a target domain by intrinsically incorporating the Wasserstein metric, which provides a natural way to measure discrepancies between probability distributions.

## 1.1 Contributions

The primary contributions of this work are summarized as follows:

**Methodology.** We propose a novel transfer learning framework for regression models with distributional outputs, addressing the challenges inherent in the Wasserstein space, which lacks a conventional linear structure. Our framework includes an efficient algorithm for cases where the informative subset of source domains is known, and a data-driven algorithm for scenarios where the subset is unknown. To the best of our knowledge, this is the first comprehensive transfer learning approach specifically designed for regression models with outputs residing in the Wasserstein space.

**Theoretical analysis.** We establish the asymptotic convergence rates for the WaTL algorithm in both the case where the informative set is known and the more challenging scenario where it must be estimated. In the latter case, we also prove that the informative set can be consistently identified. In both settings, we demonstrate that WaTL effectively improves model performance on the target data by leveraging information from the source domain. The proofs rely heavily on empirical process theory and a careful analysis of the covariate structure. Our key theoretical results extend beyond responses lying in the Wasserstein space, offering potential applications to other complex outputs.

**Simulation studies and real-world applications.** We evaluate WaTL through simulations and real data applications, demonstrating its effectiveness in improving target model performance by leveraging source domain information. The benefits become more pronounced with larger source sample sizes, underscoring its ability to harness transferable knowledge.

## 1.2 Related Work

**Transfer learning.** Transfer learning aims to improve performance in a target population by leveraging information from a related source population and has seen wide application across domains [e.g., 16, 15, 8, 41, 40]. Recent theoretical developments have focused on regression in Euclidean settings, including high-dimensional linear [21] and generalized linear models [31], nonparametric regression [6, 22], and function mean estimation from discretely sampled data [5]. In parallel, optimal transport has been used to measure distributional shifts for domain adaptation [10, 29]. However, to the best of our knowledge, no existing work has investigated transfer learning in regression models where outputs are probability distributions residing in the Wasserstein space. This represents a significant gap in the literature, highlighting the need for novel methodologies that address this challenging yet important setting.

**Distributional data analysis.** The increasing prevalence of data where distributions serve as fundamental units of observation has spurred the development of distributional data analysis [27]. Recent advancements in this field include geodesic principal component analysis in the Wasserstein space [1], autoregressive models for time series of distributions [39, 44], and distribution-on-distribution regression [14, 7]. Leveraging the Wasserstein metric, regression models with distributional outputs and Euclidean inputs can be viewed as a special case of Fréchet regression [26], which extends linear and local linear regression to outputs residing in general metric spaces. In practical scenarios where only finite samples from the unknown distributional output are available, empirical measures have been utilized as substitutes for the unobservable distributions in regression models [43].

# 2 Preliminaries

## 2.1 Notations

Let $L^2(0,1)$ be the space of square-integrable functions over the interval $(0,1)$, with the associated $L^2$ norm and metric denoted by $\|\cdot\|_2$ and $d_{L^2}$, respectively. To be specific, $\|g\|_2 = (\int_0^1 g^2(z)dz)^{1/2}$ and $d_{L^2}(g_1, g_2) = \|g_1 - g_2\|_2$. For a vector $Z$, $\|Z\|$ denotes the Euclidean norm. Given a matrix $\Sigma$, we define its spectrum as the set of its singular val-

ues. For a sub-Gaussian random vector $X$, we define the sub-Gaussian norm as $\|X\|_{\Psi_2} := \sup_{\|v\|=1} \inf \left\{ t > 0 : E(e^{\langle X,v\rangle^2/t^2}) \leq 2 \right\}$.

We write $a_n \lesssim b_n$ if there exists a positive constant $C$ such that $a_n \leq C b_n$ when $n$ is large enough and $a_n \asymp b_n$ if $a_n \lesssim b_n$ and $b_n \lesssim a_n$. The notation $a_n = O_p(b_n)$ implies that $P(|a_n/b_n| \leq C) \to 1$ for some constant $C > 0$, while $a_n = o_p(b_n)$ implies that $P(|a_n/b_n| > c) \to 0$ for any constant $c > 0$. Superscripts typically indicate different data sources, while subscripts distinguish individual samples from the same source.

## 2.2 Wasserstein Space

Let $\mathcal{W}$ denote the space of probability distributions on $\mathbb{R}$ with finite second moments, equipped with the 2-Wasserstein, or simply Wasserstein, metric. For two distributions $\mu_1, \mu_2 \in \mathcal{W}$, the Wasserstein metric is given by $d_{\mathcal{W}}^2(\mu_1, \mu_2) = \inf_{\pi \in \Pi(\mu_1, \mu_2)} \int_{\mathbb{R}\times\mathbb{R}} |s-t|^2 \, d\pi(s,t)$, where $\Pi(\mu_1, \mu_2)$ denotes the set of all joint distributions with marginals $\mu_1$ and $\mu_2$ [18]. For a probability measure $\mu \in \mathcal{W}$ with cumulative distribution function $F_\mu$, we define the quantile function $F_\mu^{-1}$ as the left-continuous inverse of $F_\mu$, such that $F_\mu^{-1}(u) = \inf\{t \in \mathbb{R} | F_\mu(t) \geq u\}$, for $u \in (0,1)$. It has been established [36] that the Wasserstein metric can be expressed as the $L^2$ metric between quantile functions:

$$d_{\mathcal{W}}^2(\mu_1, \mu_2) = \int_0^1 \left\{ F_{\mu_1}^{-1}(u) - F_{\mu_2}^{-1}(u) \right\}^2 du. \tag{1}$$

The space $\mathcal{W}$, endowed with the Wasserstein metric, forms a complete and separable metric space, commonly known as the Wasserstein space [36].

Assuming $E\{d_{\mathcal{W}}^2(\nu, \mu)\} < \infty$ for all $\mu \in \mathcal{W}$, the Fréchet mean [13] of a random distribution $\nu \in \mathcal{W}$ is given by $\nu_\oplus = \arg\min_{\mu \in \mathcal{W}} E\{d_{\mathcal{W}}^2(\nu, \mu)\}$. Since the Wasserstein space $\mathcal{W}$ is a Hadamard space [20], the Fréchet mean is well-defined and unique. Moreover, from (1), it follows that the quantile function of the Fréchet mean, denoted as $F_{\nu_\oplus}^{-1}$, satisfies $F_{\nu_\oplus}^{-1}(u) = E\{F_\nu^{-1}(u)\}, u \in (0,1)$.

## 2.3 Fréchet Regression

Consider a random pair $(X, \nu)$ with joint distribution $\mathcal{F}$ on the product space $\mathbb{R}^p \times \mathcal{W}$. Let $X$ have mean $\theta = E(X)$ and covariance matrix $\Sigma = \text{Var}(X)$, where $\Sigma$ is assumed to be positive definite. To establish a regression framework for predicting the distributional response $\nu$ from the covariate $X$, we employ the Fréchet regression model, which extends multiple linear regression and local linear regression to scenarios where responses reside in a metric space [26]. The Fréchet regression function is defined as the conditional Fréchet mean of $\nu$ given $X = x$,

$$m(x) = \arg\min_{\mu \in \mathcal{W}} E\{d_{\mathcal{W}}^2(\nu, \mu)|X = x\}.$$

For a detailed exposition of Fréchet regression, we refer the reader to [26]. Given $n$ independent realizations $\{(X_i, \nu_i)\}_{i=1}^n$, we define the empirical mean and covariance of $X$ as $\overline{X} = \frac{1}{n}\sum_{i=1}^n X_i$ and $\widehat{\Sigma} = \frac{1}{n}\sum_{i=1}^n (X_i - \overline{X})(X_i - \overline{X})^{\mathrm{T}}$.

The global Fréchet regression extends classical multiple linear regression and estimates the conditional Fréchet mean as

$$m_G(x) = \arg\min_{\mu \in \mathcal{W}} E\{s_G(x)d_{\mathcal{W}}^2(\nu, \mu)\},$$

where the weight function is given by $s_G(x) = 1 + (X - \theta)^{\mathrm{T}}\Sigma^{-1}(x - \theta)$. The empirical estimator is formulated as

$$\widehat{m}_G(x) = \arg\min_{\mu \in \mathcal{W}} \frac{1}{n}\sum_{i=1}^n s_{iG}(x)d_{\mathcal{W}}^2(\nu_i, \mu),$$

where $s_{iG}(x) = 1 + (X_i - \overline{X})^{\mathrm{T}}\widehat{\Sigma}^{-1}(x - \overline{X})$.

Similarly, local Fréchet regression extends classical local linear regression to settings with metric space-valued outputs. In the case of a scalar predictor $X \in \mathbb{R}$, the local Fréchet regression function is

$$m_{L,h}(x) = \arg\min_{\mu \in \mathcal{W}} E\{s_L(x, h)d_{\mathcal{W}}^2(\nu, \mu)\},$$

where the weight function is $s_L(x, h) = K_h(X - x)\{u_2 - u_1(X - x)\}/\sigma_0^2$, with $u_j = E\{K_h(X - x)(X - x)^j\}$, $j = 0, 1, 2$, and $\sigma_0^2 = u_0 u_2 - u_1^2$. Here, $K_h(\cdot) = h^{-1} K(\cdot/h)$ is a kernel function with bandwidth $h$. The empirical version is given by

$$\widehat{m}_{L,h}(x) = \arg\min_{\mu \in \mathcal{W}} \frac{1}{n} \sum_{i=1}^{n} s_{iL}(x, h) d_{\mathcal{W}}^2(\nu_i, \mu),$$

where $s_{iL}(x, h) = K_h(X_i - x)\{\widehat{u}_2 - \widehat{u}_1(X_i - x)\}/\widehat{\sigma}_0^2$, with $\widehat{u}_j = n^{-1} \sum_{i=1}^{n} K_h(X_i - x)(X_i - x)^j$, $j = 0, 1, 2$, and $\widehat{\sigma}_0^2 = \widehat{u}_0 \widehat{u}_2 - \widehat{u}_1^2$.

# 3 Methodology

## 3.1 Setup

We consider a transfer learning problem where target data $\{(X_i^{(0)}, \nu_i^{(0)})\}_{i=1}^{n_0}$ are sampled independently from the target population $(X^{(0)}, \nu^{(0)}) \sim \mathcal{F}_0$, and source data $\{(X_i^{(k)}, \nu_i^{(k)})\}_{i=1}^{n_k}$ are sampled independently from the source population $(X^{(k)}, \nu^{(k)}) \sim \mathcal{F}_k$, for $k = 1, \ldots, K$. The goal is to estimate the target model using both the target data and source data from $K$ related studies.

For $k = 0, \ldots, K$, assume $X^{(k)}$ has mean $\theta_k$ and covariance $\Sigma_k$, with $\Sigma_k$ positive definite. Define the empirical mean and covariance of $\{X_i^{(k)}\}_{i=1}^{n_k}$ as $\overline{X}_k = n_k^{-1} \sum_{i=1}^{n_k} X_i^{(k)}$ and $\widehat{\Sigma}_k = \frac{1}{n_k} \sum_{i=1}^{n_k} (X_i^{(k)} - \overline{X}_k)(X_i^{(k)} - \overline{X}_k)^{\mathrm{T}}$. For a fixed $x \in \mathbb{R}^p$, the weight function is $s_G^{(k)}(x) = 1 + (X^{(k)} - \theta_k)^{\mathrm{T}} \Sigma_k^{-1}(x - \theta_k)$, with the sample version $s_{iG}^{(k)}(x) = 1 + (X_i^{(k)} - \overline{X}_k)^{\mathrm{T}} \widehat{\Sigma}_k^{-1}(x - \overline{X}_k)$. The target regression function for a given $x \in \mathbb{R}^p$ is then $m_G^{(0)}(x) = \arg\min_{\mu \in \mathcal{W}} E\{s_G^{(0)}(x) d_{\mathcal{W}}^2(\nu^{(0)}, \mu)\}$. In the following, we present details on transfer learning for global Fréchet regression, where the key difference in the local Fréchet regression setting is the use of a different weight function. The technical details for transfer learning in local Fréchet regression are therefore deferred to Appendix D.

The set of informative auxiliary samples (informative set) consists of sources sufficiently similar to the target data. Formally, the informative set is defined as $\mathcal{A}_\psi = \{1 \leq k \leq K : \|f^{(0)}(x) - f^{(k)}(x)\|_2 \leq \psi\}$ for some $\psi > 0$, where $f^{(k)}(x) = E\{s_G^{(k)}(x) F_{\nu^{(k)}}^{-1}\}$. For simplicity, let $n_{\mathcal{A}} = \sum_{k=1}^{K} n_k$.

## 3.2 Wasserstein Transfer Learning

We propose the Wasserstein Transfer Learning (WaTL) algorithm, which combines information from source datasets under the assumption that all source data are *informative enough*. This assumption implies that the discrepancies between the source and target are small enough to enhance estimation compared to using only the target. When this condition is met for all source datasets, the informative set is given by $\mathcal{A}_\psi = \{1, \ldots, K\}$. The detailed steps of WaTL are presented in Algorithm 1.

---

**Algorithm 1** Wasserstein Transfer Learning (WaTL)

---

**Input:** Target and source data $\{(x_i^{(0)}, \nu_i^{(0)})\}_{i=1}^{n_0} \cup \left( \cup_{1 \leq k \leq K} \{(x_i^{(k)}, \nu_i^{(k)})\}_{i=1}^{n_k} \right)$, regularization parameter $\lambda$, and query point $x \in \mathbb{R}^p$.

**Output:** Target estimator $\widehat{m}_G^{(0)}(x)$.

1: Weighted auxiliary estimator: $\widehat{f}(x) = \frac{1}{n_0 + n_{\mathcal{A}}} \sum_{k=0}^{K} n_k \widehat{f}^{(k)}(x)$, where $\widehat{f}^{(k)}(x) = n_k^{-1} \sum_{i=1}^{n_k} s_{iG}^{(k)}(x) F_{\nu_i^{(k)}}^{-1}$.

2: Bias correction using target data: $\widehat{f}_0(x) = \arg\min_{g \in L^2(0,1)} \frac{1}{n_0} \sum_{i=1}^{n_0} s_{iG}^{(0)}(x)\|F_{\nu_i^{(0)}}^{-1} - g\|_2^2 + \lambda\|g - \widehat{f}(x)\|_2$.

3: Projection to Wasserstein space: $\widehat{m}_G^{(0)}(x) = \arg\min_{\mu \in \mathcal{W}} \left\| F_\mu^{-1} - \widehat{f}_0(x) \right\|_2$.

---

In Step 1, the initial estimate $\widehat{f}$ aggregates information from both the target and source, weighted by their respective sample sizes. While this step incorporates valuable auxiliary information, the resulting

---

**Algorithm 2** Adaptive Wasserstein Transfer Learning (AWaTL)

---

**Input:** Target and source data $\{(x_i^{(0)}, \nu_i^{(0)})\}_{i=1}^{n_0} \cup \left( \cup_{1 \leq k \leq K} \{(x_i^{(k)}, \nu_i^{(k)})\}_{i=1}^{n_k} \right)$, regularization parameter $\lambda$, prespecified number of informative sources $L$, and query point $x \in \mathbb{R}^p$.

**Output:** Target estimator $\widehat{m}_G^{(0)}(x)$.

1: Compute discrepancy scores. For each source dataset $k = 1, \ldots, K$, compute the empirical discrepancy: $\widehat{\psi}_k = \|\widehat{f}^{(0)}(x) - \widehat{f}^{(k)}(x)\|_2$, where $\widehat{f}^{(k)}(x) = n_k^{-1} \sum_{i=1}^{n_k} s_{iG}^{(k)}(x) F_{\nu_i^{(k)}}^{-1}$. Construct the adaptive informative set by selecting the $L$ smallest discrepancy scores $\widehat{\mathcal{A}} = \{ k : 1 \leq k \leq K$ and $\widehat{\psi}_k$ is among the $L$ smallest values $\}$..

2: Weighted auxiliary estimator: $\widehat{f}(x) = \frac{1}{\sum_{k \in \widehat{\mathcal{A}} \cup \{0\}} n_k} \sum_{k \in \widehat{\mathcal{A}} \cup \{0\}} n_k \widehat{f}^{(k)}(x)$.

3: Bias correction using target data: $\widehat{f}_0(x) = \arg\min_{g \in L^2(0,1)} \frac{1}{n_0} \sum_{i=1}^{n_0} s_{iG}^{(0)}(x) \|F_{\nu_i^{(0)}}^{-1} - g\|_2^2 + \lambda \|g - \widehat{f}(x)\|_2$.

4: Projection to Wasserstein space: $\widehat{m}_G^{(0)}(x) = \arg\min_{\mu \in \mathcal{W}} \left\| F_\mu^{-1} - \widehat{f}_0(x) \right\|_2$.

---

estimate may be biased due to distributional differences between the target and source populations. In Step 2, the bias in $\widehat{f}$ is corrected by focusing on the target data. The regularization term $\lambda \|g - \widehat{f}(x)\|_2$ ensures a balance between target-specific precision and auxiliary-informed robustness. Theoretical guidelines for selecting $\lambda$ are provided in Theorem 2. The final step projects the corrected estimate $\widehat{f}_0$ onto the Wasserstein space, ensuring the output $\widehat{m}_G^{(0)}(x)$ respects the intrinsic geometry of $\mathcal{W}$. This projection exists and is unique because $\mathcal{W}$ is a closed and convex subset of $L^2(0,1)$.

### 3.3 Adaptive Selection of Informative Sources

In many practical scenarios, the assumption that all source datasets belong to the informative set $\mathcal{A}_\psi$ may not hold. To address this, we extend WaTL with an adaptive selection procedure to identify the informative set. The discrepancy for each source dataset $k$ is defined as $\psi_k = \|f^{(0)}(x) - f^{(k)}(x)\|_2$, which measures the distance between the target distribution and the auxiliary distribution. Since $f^{(0)}(x)$ and $f^{(k)}(x)$ are unknown, we compute an empirical estimate $\widehat{\psi}_k$ for $\psi_k$, which is used to adaptively estimate the informative set $\mathcal{A}_\psi$. To implement this approach, an additional input parameter $L$, which specifies the approximate number of informative sources, is required. In practice, $L$ can be treated as a tuning parameter and selected through cross-validation or other model selection techniques. The full procedure is formalized in Algorithm 2.

The proposed algorithm adaptively identifies the informative set $\widehat{\mathcal{A}}$ in Step 1 by evaluating the empirical discrepancy scores $\widehat{\psi}_k$. The selected set is then used to compute the weighted auxiliary estimator in Step 2, ensuring that only the most relevant source datasets contribute to the final target estimator. Steps 3 and 4 follow the same bias correction and projection procedures as described in Section 3.2. This adaptive approach enhances the robustness of WaTL by excluding irrelevant or highly dissimilar source datasets.

## 4 Theory

In this section, we establish the theoretical guarantees of the proposed WaTL and AWaTL algorithms using techniques from empirical process theory [34]. For WaTL, we present the following lemma, which characterizes the convergence rate for each term contributing to the weighted auxiliary estimator $\widehat{f}(x)$ computed in Step 1.

**Condition 1.** *For $k = 0, \ldots, K$, the covariate $X^{(k)}$ is sub-Gaussian with $\|X^{(k)}\|_{\Psi_2} \in [\sigma_1, \sigma_2]$, the mean vector satisfies $\|\theta_k\| \leq R_1$, and the spectrum of the covariance matrix $\Sigma_k$ lies within the interval $[R_2, R_3]$. Moreover, $\nu^{(k)}$ is supported on a bounded interval.*

**Lemma 1.** *Let* $\widehat{f}^{(k)}(x) = n_k^{-1} \sum_{i=1}^{n_k} s_{iG}^{(k)}(x) F_{\nu_i^{(k)}}^{-1}$ *and its population counterpart be defined as* $f^{(k)}(x) = E\{s_G^{(k)}(x) F_{\nu^{(k)}}^{-1}\}$ *for* $k = 0, \ldots, K$. *Then under Condition 1,* $\|\widehat{f}^{(k)}(x) - f^{(k)}(x)\|_2 = O_p(n_k^{-1/2})$.

To derive the convergence rate of $\widehat{f}(x)$, we rely on the following condition.

**Condition 2.** *There exist positive constants* $C_1, C_2, C_3, C_4$ *such that*

$$\sum_{k=0}^{K} 2e^{-C_1 \frac{4}{R_3^2} n_k} + C_2 e^{-C_3 \left( \frac{2}{R_3} - \sqrt{\frac{C_4}{n_k}} \right) n_k} = o(1),$$

*where* $R_3$ *is as in Condition 1. In addition,*

$$\frac{\sqrt{n_0 + n_{\mathcal{A}}}}{\min_{1 \leq k \leq K} n_k} = o(1), \quad \frac{\sqrt{n_0 + n_{\mathcal{A}}}}{n_0} = o(1).$$

*Remark* 1. These conditions are typically satisfied in practice as they are not overly restrictive. Condition 1 requires that covariates and covariance matrices are bounded in a specific way, which is standard in the transfer learning literature and generally holds in real-world scenarios [5]. In particular, this assumption is common when dealing with high-dimensional data where regularization is necessary [21]. Condition 2 assumes that the number of samples is significantly larger than the number of sources $K$, which is reasonable since $K$ is usually fixed in practical settings, and we often have sufficient source data compared to target data. In practice, Condition 2 may be slightly violated if there exists a source $k$ with a relatively small $n_k$, such that $\sqrt{n_0 + n_{\mathcal{A}}}/n_k$ is not $o(1)$. In such cases, the $k$th source can simply be excluded from Step 1 of the WaTL algorithm.

**Theorem 1.** *Suppose Conditions 1 and 2 hold. Then, for the WaTL algorithm, it holds for a fixed* $x \in \mathbb{R}^p$ *that*

$$\|\widehat{f}(x) - f(x)\|_2 = O_p\left( \frac{\sum_{k=0}^{K} \sqrt{n_k}}{n_0 + n_{\mathcal{A}}} + (n_0 + n_{\mathcal{A}})^{-1/2} \right),$$

*where* $f(x) = (n_0 + n_{\mathcal{A}})^{-1} \sum_{k=0}^{K} n_k f^{(k)}(x)$.

The proof of Theorem 1 involves a detailed analysis of the sample covariance matrix and leverages M-estimation techniques within the framework of empirical process theory. The result extends beyond responses in the Wasserstein space, applying to other metric spaces that meet mild regularity conditions. Consequently, Theorem 1 provides a versatile framework that can be applied to transfer learning in regression models with responses such as networks [42], symmetric positive-definite matrices [25], or trees [2]. The following theorem establishes the convergence rate for the estimated regression function $\widehat{m}_G^{(0)}(x)$ in the WaTL algorithm.

**Theorem 2.** *Assume Conditions 1 and 2 hold and the regularization parameter satisfies* $\lambda \asymp n_0^{-1/2+\epsilon}$ *for some* $\epsilon > 0$. *Then, for the WaTL algorithm and a fixed* $x \in \mathbb{R}^p$, *it holds that*

$$d_{\mathcal{W}}^2(\widehat{m}_G^{(0)}(x), m_G^{(0)}(x)) = O_p\left( n_0^{-1/2+\epsilon} \left( \psi + \frac{\sum_{k=0}^{K} \sqrt{n_k}}{n_0 + n_{\mathcal{A}}} + (n_0 + n_{\mathcal{A}})^{-1/2} \right) \right),$$

*where* $\psi = \max_{1 \leq k \leq K} \|f^{(0)}(x) - f^{(k)}(x)\|_2$ *quantifies the maximum discrepancy between the target and source.*

Theorem 2 can be compared to the convergence rate of global Fréchet regression [26] applied solely to the target data, for which the rate is $d_{\mathcal{W}}(\widehat{m}_G^{(0)}(x), m_G^{(0)}(x)) = O_p(n_0^{-1/2})$. The WaTL algorithm achieves a faster convergence rate when there are sufficient source data and the auxiliary sources are informative enough, satisfying $\psi \lesssim n_0^{-1/2-\epsilon}$. This result highlights that knowledge transfer from auxiliary samples can significantly enhance the learning performance of the target model, provided the auxiliary sources are closely aligned with the target data.

For the AWaTL algorithm, we require the following condition.

**Condition 3.** *The regularization parameter satisfies* $\lambda \asymp n_0^{-1/2+\epsilon}$ *for some* $\epsilon > 0$ *and for some* $\epsilon' > \epsilon$, *there exists a non-empty subset* $\mathcal{A} \subset \{1, \ldots, K\}$ *such that*

$$\frac{n_*^{-1/2} + n_0^{-1/2}}{\min_{k \in \mathcal{A}^C} \psi_k} = o(1), \quad \frac{\max_{k \in \mathcal{A}} \psi_k}{n_*^{-1/2} + n_0^{-1/2-\epsilon'}} = O(1),$$

*where $\mathcal{A}^C = \{1, \ldots, K\} \backslash \mathcal{A}$, $n_* = \min_{1 \leq k \leq K} n_k$ and $\psi_k = \|f^{(0)}(x) - f^{(k)}(x)\|_2$.*

*Remark* 2. We allow $\mathcal{A}$ to be the whole set $\{1, \ldots, K\}$, in which case the condition becomes

$$\frac{\max_{1 \leq k \leq K} \psi_k}{n_*^{-1/2} + n_0^{-1/2-\epsilon'}} = O(1).$$

Besides, if $\mathcal{A}$ exists, then it is unique since

$$\{1 \leq k \leq K : \frac{n_*^{-1/2} + n_0^{-1/2}}{\psi_k} = o(1)\} \cap \{1 \leq k \leq K : \frac{\psi_k}{n_*^{-1/2} + n_0^{-1/2-\epsilon'}} = O(1)\} = \emptyset.$$

Condition 3 ensures that the source datasets can be effectively partitioned into two groups: informative ones and those sufficiently different from the target data. This separation guarantees that the informative set can be accurately identified. Under this condition, by setting the number of informative sources to $L = |\mathcal{A}|$, we establish the following rate of convergence for the AWaTL algorithm. In practice, $L$ can be decided by cross-validation.

**Theorem 3.** *Under Condition 3 and the conditions of Theorem 2, for the AWaTL algorithm with a fixed number of sources $K$ we have $P(\widehat{\mathcal{A}} = \mathcal{A}) \to 1$ and*

$$d_{\mathcal{W}}^2(\widehat{m}_G^{(0)}(x), m_G^{(0)}(x)) = O_p\Big(n_0^{-1/2+\epsilon}\big(\max_{k \in \mathcal{A}} \psi_k + \frac{\sum_{k \in \mathcal{A} \cup \{0\}} \sqrt{n_k}}{\sum_{k \in \mathcal{A} \cup \{0\}} n_k} + \big(\sum_{k \in \mathcal{A} \cup \{0\}} n_k\big)^{-1/2}\big)\Big).$$

The convergence rate of the AWaTL algorithm simplifies to $n_0^{-1-\epsilon'+\epsilon}$ when the informative source data is sufficiently large. This rate surpasses that of global Fréchet regression [26] applied solely to the target data, offering a theoretical guarantee that AWaTL effectively mitigates negative transfer by selectively integrating relevant auxiliary information.

While the above analysis assumes that each probability distribution is fully observed, an assumption commonly adopted in the distributional data literature [26, 7], real-world applications often provide only independent samples drawn from underlying distributions. In such cases, this limitation can be overcome by replacing unobservable distributions with their empirical counterparts, constructed from sample observations [43]. Additional details and theoretical justification for this extension are provided in Appendix E.

## 5 Numerical Experiments

In this section, we evaluate the performance of the proposed WaTL algorithm, alongside two baseline approaches: the global Fréchet regression using only target data (Only Target) and using only source data (Only Source). Consider $K = 5$ source sites. The data are generated as follows. For the target population, we sample $X^{(0)} \sim \text{U}(0,1)$ and generate the response distribution, represented by its quantile function, as

$$F_{\nu^{(0)}}^{-1}(u) = w^{(0)}(1-u)u + (1 - X^{(0)})u + X^{(0)} F_{Z^{(0)}}^{-1}(u), \quad u \in (0,1),$$

where $Z^{(0)} \sim N(0.5, 1)|_{(0,1)}$ and $w^{(0)} \sim N(0,1)|_{(-0.5,0.5)}$. Here, $N(\mu, \sigma^2)|_{(a,b)}$ denotes a normal distribution with mean $\mu$ and variance $\sigma^2$, truncated to the interval $(a, b)$. For source populations, we define $\psi_k = 0.1k$ for $k = 1, \ldots, K$, and generate $X^{(k)} \sim \text{U}(0,1)$. The corresponding response distribution is generated as

$$F_{\nu^{(k)}}^{-1}(u) = w^{(k)}(1-u)u + (1 - X^{(k)})u + X^{(k)} F_{Z^{(k)}}^{-1}(u), \quad u \in (0,1),$$

where $Z^{(k)} \sim N(0.5, 1-\psi_k)|_{(0,1)}$ and $w^{(k)} \sim N(0,1)|_{(-0.5,0.5)}$. Consequently, for each predictor $x$, the true regression function is $m_G^{(k)}(x) = (1-x)u + x F_{Z^{(k)}}^{-1}(u)$, for $k = 0, 1, \ldots, K$.

We vary the target sample size $n_0$ from 200 to 800, while the source sample size is set as $n_k = k\tau$, where $\tau \in \{100, 200\}$ and $k = 1, \ldots, K$. The regularization parameter $\lambda$ in Algorithm 1 is selected via five-fold cross-validation, ranging from 0 to 3 in increments of 0.1. To evaluate performance, we sample 100 predictors uniformly from the target distribution. Using Algorithm 1, we compute

$\widehat{m}_G^{(0)}(x)$ and compare it with the corresponding estimates obtained from global Fréchet regression using only target or source data. Performance is assessed using the root mean squared prediction risk RMSPR $= \sqrt{\frac{1}{100} \sum_{i=1}^{100} d_{\mathcal{W}}^2 (\widehat{m}_G^{(0)}(x_i), m_G^{(0)}(x_i))}$, where $x_i$ denotes the sampled predictor, $\widehat{m}_G^{(0)}(x_i)$ is the estimated function, and $m_G^{(0)}(x_i)$ represents the ground truth. To ensure robustness, we repeat the simulation 50 times and report the average RMSPR, as shown in Figure 1(a).

As shown in Figure 1(a), WaTL consistently outperforms global Fréchet regression trained solely on target or source data. When the target sample size $n_0$ is small, the Only Target method exhibits a high RMSPR due to the instability of models trained on limited data. In contrast, WaTL significantly reduces RMSPR by effectively incorporating auxiliary information from the source domain. As $n_0$ increases, the performance of Only Target improves and gradually approaches that of WaTL, which is expected as larger sample sizes lead to more stable and accurate estimators. Nevertheless, WaTL maintains a consistent advantage across all $n_0$, suggesting that it leverages complementary information from the source. The performance of the Only Source estimator remains nearly unchanged across different $n_0$ values, as it does not benefit from additional target data.

Comparing the two panels of Figure 1(a), we also observe that WaTL improves as the source sample size increases, confirming its ability to effectively integrate information from source domains. This demonstrates the benefit of multi-source transfer learning, where WaTL balances knowledge from both target and source domains to achieve improved prediction.

To better understand when negative transfer may occur, we conduct an ablation study with $K = 1$ source and vary the similarity parameter $\psi_1$ from 0.01 to 1 in increments of 0.01, with $n_0 = 100$ and $n_1 = 200$. Our method outperforms the Only Target approach when $\psi_1 < 0.9$, while for $\psi_1 \geq 0.9$, the Only Target method becomes preferable. This confirms that negative transfer arises when the source is too dissimilar to the target.

We further evaluate the effectiveness of AWaTL in selecting informative sources. In this experiment, we set $L = 2$ and $n_k = 100$ for $k = 0, \ldots, 5$. The similarity parameters are specified as $\psi_k = 0.1$ for $k = 1, 2$ (informative sources), and $\psi_k = \psi$, increasing from 0.2 to 1 in increments of 0.1, for $k = 3, 4, 5$ (uninformative sources). Each configuration is repeated 100 times, and the corresponding selection rates are reported in Figure 1(b). The results show that AWaTL successfully identifies informative sources, with selection rates for sources 1 and 2 rapidly increasing and reaching perfect accuracy once $\psi > 0.6$. These findings demonstrate the robustness of AWaTL in distinguishing useful sources under varying similarity levels.

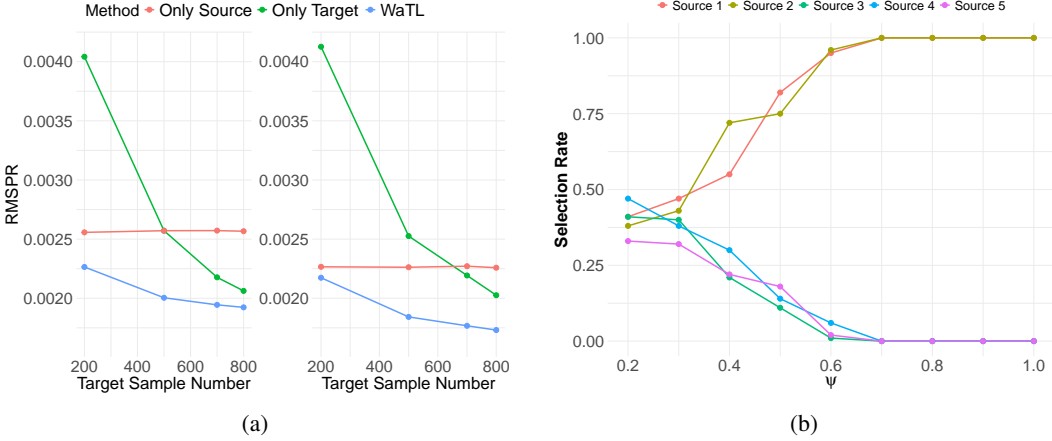

Figure 1: (a) Root mean squared prediction risk (RMSPR) of WaTL, only Source, and Only Target methods under varying target sample sizes, with source sample sizes $\tau = 100$ (left) and $\tau = 200$ (right); (b) Selection rate of each source site as $\psi$ increases.

# 6 Real-world Applications

We evaluate the WaTL algorithm using data from the National Health and Nutrition Examination Survey (NHANES) 2005–2006[3], focusing on modeling the distribution of physical activity intensity. NHANES is a large-scale health survey in the United States that combines interviews with physical examinations to assess the health and nutrition of both adults and children. The dataset includes extensive demographic, socioeconomic, dietary, and medical assessments, providing a comprehensive resource for health-related research.

During the 2005–2006 NHANES cycle, participants aged 6 and older wore an ActiGraph 7164 accelerometer on their right hip for seven days, recording physical activity intensity in 1-minute epochs. Participants were instructed to remove the device during water-based activities and sleep. The device measured counts per minute (CPM), ranging from 0 to 32767, capturing variations in activity levels throughout the monitoring period.

Since female and male participants exhibit distinct physical activity patterns [12], we analyze them separately. Physical activity intensity is influenced by multiple factors, and we consider body mass index (BMI) and age as key predictors [19, 9]. To accommodate potential nonlinear relationships, we implement local Fréchet regression within the WaTL algorithm, treating the distribution of physical activity intensity as the response and BMI and age as predictors.

Following the data preprocessing steps in [23], we remove unreliable observations per NHANES protocols. For each participant, we exclude activity counts above 1000 CPM or equal to zero (as zeros may correspond to various low-activity states such as sleep or swimming). Participants with fewer than 100 valid observations or missing BMI, age, or gender information are also excluded. The remaining activity counts over seven days are concatenated to form the distribution of each participant's activity intensity.

To evaluate the WaTL, we set White, Mexican Americans, and other Hispanic individuals as sources and Black as the target. For females, the source data include 1308 White people, 884 Mexican Americans, and 108 Other Hispanic individuals. For males, the source data include 1232 White participants, 805 Mexican Americans, and 92 Other Hispanic individuals. We set 200 Black participants as the target data for both genders. During evaluating, we perform five-fold cross-validation, using four folds for training and one for testing, cycling through each fold. For comparison, we also apply local Fréchet regression using only the target data. The results, summarized in Figure 2(a), show that WaTL improves performance over local Fréchet regression for both females and males, demonstrating its ability to leverage information from other demographic groups to enhance modeling for Black participants.

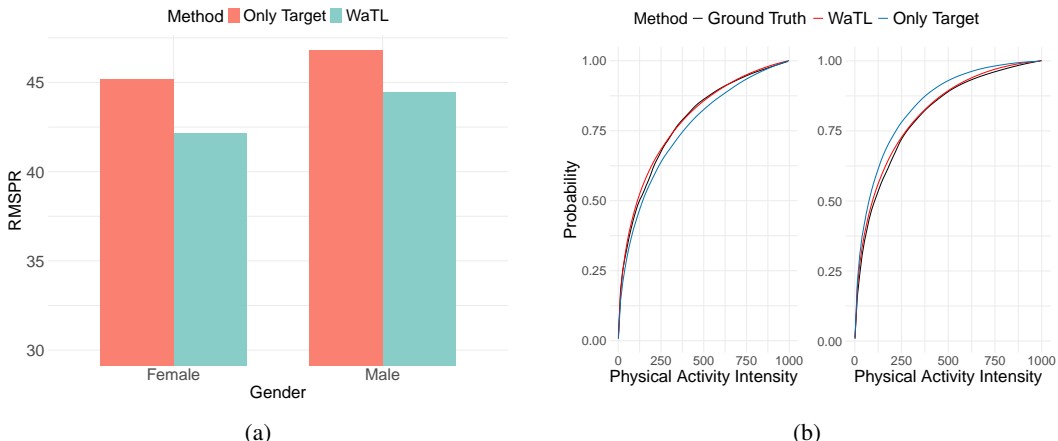

(a)                                    (b)

Figure 2: (a) Root mean squared prediction risk (RMSPR) of WaTL and Only Target methods for females and males, evaluated using five-fold cross-validation; (b) Cumulative distribution function of physical activity levels for one selected female (left) and one selected male (right), along with estimates from WaTL and Only Target methods.

[3]https://wwwn.cdc.gov/nchs/nhanes/ContinuousNhanes/Default.aspx?BeginYear=2005

To further illustrate the effectiveness of WaTL, we visualize the cumulative distribution function of physical activity levels for one female participant and one male participant in Figure 2(b), along with estimates from WaTL and local Fréchet regression, using only the target data. The results indicate that WaTL provides a better fit to the true distribution, outperforming the estimate obtained using only the target data. We further evaluate the robustness of WaTL on a human mortality dataset in Appendix A, where WaTL continues to demonstrate strong performance.

## 7    Conclusion

We introduce Wasserstein transfer learning and its adaptive variant for scenarios where the informative set is unknown, addressing the challenges posed by the lack of linear operations in the Wasserstein space. By leveraging the Wasserstein metric, the proposed algorithm accounts for the non-Euclidean structure of distributional outputs, ensuring compatibility with the intrinsic geometry of the Wasserstein space. Supported by rigorous theoretical guarantees, the framework demonstrates improved estimation performance compared to methods that rely solely on target data.

This paper focuses on univariate distributions, which arise frequently in real-world applications such as the NHANES and human mortality studies discussed in Section 6. The proposed framework, however, is not limited to the univariate case and can be extended to multivariate distributions by incorporating metrics such as the Sinkhorn [11] or sliced Wasserstein distance [4]. Extending the methodology to higher dimensions is an exciting future direction that entails addressing both computational and theoretical challenges specific to high-dimensional Wasserstein spaces.

## Acknowledgments and Disclosure of Funding

We thank the Area Chair and the reviewers for their constructive feedback. Doudou Zhou was supported by the NUS Start-Up Grant (A-0009985-00-00) and the MOE AcRF Tier 1 Grant (A-8003569-00-00).

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

# A Human Mortality Data

We further assess WaTL using the age-at-death distributions from 162 countries in 2015, compiled from the United Nations Databases[4] and the UN World Population Prospects 2022[5]. The dataset provides country-level age-specific death counts, which we convert into smooth age-at-death densities using local linear smoothing; see Figure 3 (a) for an illustration. For this analysis, we define the 24 developed countries as the target site and the remaining 138 developing countries as the source site.

We evaluate the performance of WaTL against several key baselines on this dataset. These include models trained only on the target data (`Only Target`), only on the source data (`Only Source`), and on a naive pooling of both (`Target + Source`). Furthermore, for these baselines, we compare the performance of our underlying Fréchet regression framework against Wasserstein Regression [7], another state-of-the-art method for distributional data.

Table 1: Performance and Training Time with Varying Target Sample Sizes.

| Number of Target Samples | RMSPR | Training Time (ms) |
|---|---|---|
| 14 | 0.028 | 0.598 |
| 19 | 0.025 | 0.597 |
| 24 | 0.022 | 0.694 |

The comprehensive results are presented in Figure 3 (b). The proposed WaTL method achieves the lowest Root Mean Squared Prediction Risk (RMSPR) of 0.022, demonstrating a marked improvement over all alternatives. Notably, it significantly outperforms models trained solely on the 24 target samples (RMSPR of 0.027 for Fréchet Regression and 0.025 for Wasserstein Regression) and the naive data pooling approach (RMSPR of 0.033). This highlights that simply combining datasets is insufficient to overcome the domain shift between developing and developed countries, validating the necessity of our bias-correction mechanism.

To further assess WaTL's robustness and practicality, especially in common scenarios with limited target data, we analyze its performance with a varying number of target samples. As shown in Table 1, the model's predictive accuracy consistently improves as the target sample size increases from 14 to 24. It is also worth noting that the method maintains exceptional computational efficiency, with training times remaining under one millisecond. These findings confirm that WaTL is not only highly accurate but also a robust and efficient solution for real-world demographic studies where target data can be scarce.

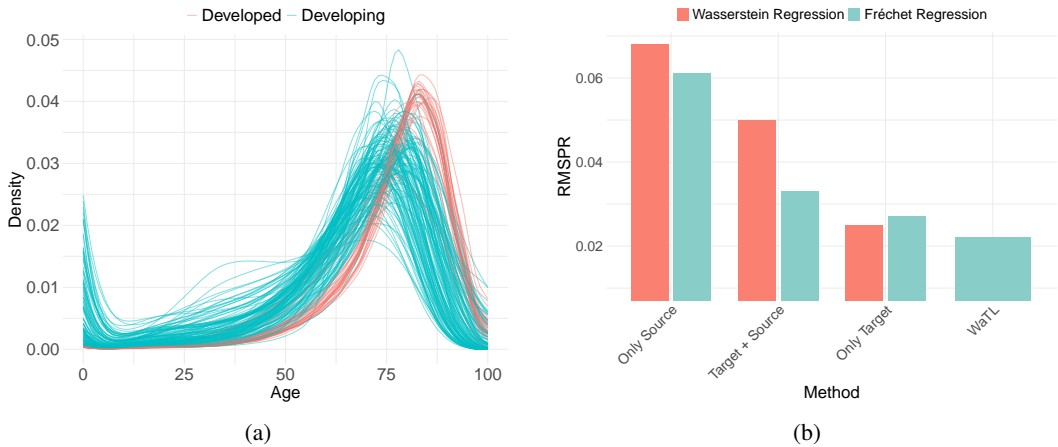

(a)  (b)

Figure 3: (a) Age-at-death densities of developed and developing countries; (b) Root mean squared prediction risk (RMSPR) of WaTL and Only Target methods for human mortality data.

---

[4] https://data.un.org/

[5] https://population.un.org/wpp/Download

## B    Additional Notations

For any $g_1, g_2 \in L^2(0,1)$, the $L^2$ inner product is defined as

$$\langle g_1, g_2 \rangle_2 = \int_0^1 g_1(z) g_2(z) dz.$$

The total variation of a function $g$ on the interval $(a, b)$ is

$$V_a^b(g) = \lim_{n \to +\infty} \sup_{a < x_1 < \ldots < x_n < b} \sum_{i=1}^n |g(x_i) - g(x_{i-1})|.$$

The space of bounded variation functions is given by

$$BV((0,1), H) = \{g \colon (0,1) \to (-H, H) | V_a^b(g) \leq H\},$$

which is equipped with the $L^2$ metric.

For a matrix $\Sigma$, let $\|\Sigma\|$ and $\|\Sigma\|_F$ denote its operator norm and Frobenius norm, respectively, and let $\gamma_i(\Sigma)$ be its $i$th smallest singular value.

We denote the characteristic function of a set $\mathcal{A}$ by $1_\mathcal{A}$ and use $|\mathcal{A}|$ to represent its cardinality. In a metric space $(\Omega, d)$, the covering number $\mathcal{N}(K, d, \epsilon)$ represents the smallest number of closed balls of radius $\epsilon$ required to cover a subset $K \subset \Omega$. The notation $\bigsqcup$ is used to denote the disjoint union of sets.

For notational simplicity, we will omit $(x)$ from $f(x)$ when the meaning is clear from the context.

## C    Proof

**Lemma 2.** *Consider $U : \mathcal{W} \times (0,1) \mapsto \mathbb{R}$, where $U(\nu, x) := F_\nu^{-1}(x)$ and $\mathcal{W} \times (0,1)$ is endowed with the product $\sigma-$algebra generated by the Borel algebra on $\mathcal{W}$ and the Borel algebra on $(0,1)$. The first Borel algebra $\mathcal{B}(\mathcal{W})$ is generated by open balls in $(\mathcal{W}, d_\mathcal{W})$ and the second $\mathcal{B}((0,1))$ is generated by Euclidean open balls. Then $U$ is measurable.*

Lemma 2 enables a simplified expression for the conditional Fréchet mean of $\nu$ given $X = x$ by applying Fubini's theorem. Specifically, we have

$$m(x) = \arg\min_{\mu \in \mathcal{W}} E\{d_\mathcal{W}^2(\nu, \mu) | X = x\}$$
$$= \arg\min_{\mu \in \mathcal{W}} E\{d_{L^2}^2(F_\nu^{-1}, F_\mu^{-1}) | X = x\}$$
$$= \arg\min_{\mu \in \mathcal{W}} d_{L^2}^2(E\{F_\nu^{-1} | X = x\}, F_\mu^{-1}),$$

where the last equality follows from Fubini's theorem, since $E\{F_\nu^{-1} | X = x\}$ is measurable. A similar argument applies to $m_G(x)$ and $\hat{m}_G(x)$.

### C.1    Proof of Lemma 2

*Proof.* Without loss of generality, It suffices to prove $\{(\nu, x) | F_\nu^{-1}(x) > 0\}$ is measurable. Note that any quantile function is left continuous, hence $\{(\nu, x) | F_\nu^{-1}(x) > 0\} = \cup_{q \in \mathbb{Q} \cap (0,1)} \{(\nu, q) | F_\nu^{-1}(q) > 0\}$.

Then without loss of generality, we suffice to prove $\{(\nu, 0.5) | F_\nu^{-1}(0.5) > 0)\}$ is measurable. Since $\mathcal{W}$ is separable, we can select a dense countable subset $K$. We then define $A = \{\nu \in K | F_\nu^{-1}(0.5) > 0\}$. Assuming $A = \{\nu_i, i \in \mathbb{N}\}$, we have $\{\nu | F_\nu^{-1}(0.5) > 0\} = \{\nu | \underline{\lim}_{i \to \infty} d_\mathcal{W}(\nu_i, \nu) = 0\} = \lim_{n \to \infty} \overline{\lim}_{i \to \infty} B_{\frac{1}{n}}(\nu_i)$ where $B_\epsilon(\nu)$ is the ball centered at $\nu$ with radius $\epsilon$. Here we remark that both left continuous and monotone increasing are used in the first equation. Hence $U$ is measurable. $\qquad\square$

## C.2 Proof of Lemma 1

*Proof.* First, it is easy to check that $f^{(k)} \in BV((0,1), H_0)$ for some $H_0 > 0$ since $f^{(k)} = E\{s_G^{(k)}(x)1_{\{s_G^{(k)}(x)>0\}}F_{\nu^{(k)}}^{-1}\} - E\{|s_G^{(k)}(x)|1_{\{s_G^{(k)}(x)<0\}}F_{\nu^{(k)}}^{-1}\}$ and $E|s_G^{(k)}(x)| < \infty$. By taking $H_0$ large enough, We also claim that $P(\widehat{f}^{(k)} \in BV((0,1), H_0)) \to 1$ since $\widehat{f}^{(k)}$ also has a similar decomposition and

$$\frac{1}{n_k}\sum_{i=1}^{n_k}|s_{iG}^{(k)}(x)| \leq \frac{1}{n_k}\sum_{i=1}^{n_k}(|s_{iG}^{(k)}(x) - s_i^{(k)}(x)| + |s_i^{(k)}(x)|)$$

$$\leq o_p(1) + H_1$$

for some $H_1 > 0$ with high probability, where $s_i^{(k)}(x) = 1 + (X_i^{(k)} - \theta_k)\Sigma_k^{-1}(x - \theta_k)$. The last inequality holds because of the result given by the proof of Theorem 2 in [26] that $\frac{1}{n_k}\sum_{i=1}^{n_k}|s_{iG}^{(k)}(x) - s_i^{(k)}(x)| = o_p(1)$ and the assumption that $X^{(k)}$ is subguassian.

To use Theorem 2 in [26], taking $\Omega$ to be $BV((0,1), H_0)$, $M(g, x)$ to be $M^{(k)}(g, x) = E\{s_G^{(k)}(x)\|F_{\nu^{(k)}}^{-1} - g\|_2^2\}$ for any $g \in BV((0,1), H_0)$, it suffices to check that the other two assumptions of the theorem:

$$\int_0^1 \sqrt{1 + \log\mathcal{N}(B_\delta(f^{(k)}) \cap BV((0,1), H_0), d_{L^2}, \delta\epsilon)}d\epsilon = O(1)$$

as $\delta \to 0$ and

$$M^{(k)}(g, x) - M^{(k)}(f^{(k)}, x) \geq \|f^{(k)} - g\|_2^2.$$

Example 19.11 in [33] gives a bound of the covering number of the space $(BV((0,1), H_0), d_{L^2})$,

$$\mathcal{N}(BV((0,1), H_0), d_{L^2}, \epsilon) \leq e^{\frac{K}{\epsilon}}$$

for some $K > 0$ where $K$ is independent of $\epsilon$.

The rest is similar to the proof of Proposition 1 in [26]. For $g \in BV((0,1), H_0)$, $B_\gamma(g)$ denotes the $L^2$ ball of radius $\gamma$ centered at $g$. Let $\mathcal{C}_\epsilon(f^{(k)}) := \{g_u : u \in U\}$ such that $|U| = |\mathcal{N}(BV((0,1), H_0) \cap B_1(f), d_{L^2}, \epsilon)| \leq e^{K\epsilon^{-1}}$ and the balls $B_\epsilon(g_u)$ covers $B_1(f) \cap BV((0,1), H_0)$. For $\delta > 0$, we define $\tilde{g}_u = f^{(k)} + \delta(g_u - f^{(k)})$. Then the balls $B_{\delta\epsilon}(\tilde{g}_u)$ covers $B_\delta(f) \cap BV((0,1), H_0)$. Hence

$$\int_0^1 \sqrt{1 + \log\mathcal{N}(B_\delta(m_G(x)) \cap BV((0,1), H_0), d_{L^2}, \delta\epsilon)}d\epsilon \leq \int_0^1 \sqrt{1 + K\epsilon^{-1}}d\epsilon \leq 1 + 2\sqrt{K} < \infty.$$

For the last assumption, just note that

$$M^{(k)}(g, x) - M^{(k)}(f^{(k)}, x) = \|f^{(k)} - g\|_2^2.$$

Then according to Theorem 2 in [26], $\|\widehat{f}^{(k)} - f^{(k)}\|_2 = O_p(n_k^{-1/2})$. $\qquad \square$

## C.3 Proof of Theorem 1

To prove Theorem 1, we first establish a lemma that quantifies the bias introduced in Step 1 of Algorithm 1. This lemma is formulated for a general metric space, making it broadly applicable and potentially useful for extending transfer learning algorithms to other metric spaces.

Here are some notations, most of which are similar to our original setting. The only difference is that we use $Y$ instead of $\nu$ to represent a random object in a general metric space $(\Omega, d)$. We define $n_* = \min_{1 \leq k \leq K} n_k$, $n_{\mathcal{A}} = \sum_{i=1}^K n_i$, $\alpha_k = n_k/(n_0 + n_{\mathcal{A}})$,

$$m_1(x) = \arg\min_{\omega \in \Omega} M^1(\omega, x), \quad M^1(\omega, x) = \sum_{k=0}^K \alpha_k E\{s_G^{(k)}(x)d^2(\omega, Y_i^{(k)})\},$$

and

$$\widehat{m}_1(x) = \arg\min_{\omega \in \Omega} M_n^1(\omega, x), \quad M_n^1(\omega, x) = \frac{1}{n_0 + n_{\mathcal{A}}}\sum_{k=0}^K\sum_{i=1}^{n_k} s_{iG}^{(k)}(x)d^2(\omega, Y_i^{(k)}).$$

We impose the following mild condition on the metric space $(\Omega, d)$, which is widely used in the literature on non-Euclidean data analysis [26]. This condition holds for various metric spaces commonly encountered in real-world applications, including the Wasserstein space, the space of networks, and the space of symmetric positive-definite matrices, among others.

**Condition 4.** *(i) $m_1(x)$ uniquely exists and $\widehat{m}_1(x)$ uniquely exists almost surely. Additionally for any $\gamma > 0$, $\inf_{d(m_1(x),\omega)>\gamma} M^1(\omega, x) > M^1(m_1(x), x)$.*

*(ii) Let $B_\delta(m_1(x)) \subset \Omega$ be the ball of radius $\delta$ centered at $m_1(x)$. Then*

$$J(\delta) := \int_0^1 \sqrt{1 + \log \mathcal{N}(B_\delta(m_1(x)), d, \delta\epsilon)} d\epsilon = O(1)$$

*as $\delta \to 0$.*

*(iii) There exist $\eta > 0$, $C > 0$ and $\beta > 1$, possibly depending on $x$, such that whenever $d(m_1(x), \omega) < \eta$, we have $M^1(\omega, x) - M_n^1(m_1(x), x) \geq C d^\beta(\omega, m_1(x))$.*

**Lemma 3.** *Under Conditions 1, 2 and 4, $d^2(\widehat{m}_1(x), m_1(x)) = O_p\left(\left(\frac{\sum_{k=0}^{K} \sqrt{n_k}}{n_0 + n_\mathcal{A}} + (n_0 + n_\mathcal{A})^{-1/2}\right)^{\frac{1}{\beta-1}}\right)$.*

### C.3.1 Proof of Lemma 3

*Proof.* Denote $V_n(\omega, x) = M_n^1(\omega, x) - M^1(\omega, x)$, and $D_i^{(k)}(\omega, x) = d^2(Y_i^{(k)}, \omega) - d^2(Y_i^{(k)}, m_1(x))$ and recall $s_i^{(k)}(x) = 1 + (X_i^{(k)} - \theta_k)\Sigma_k^{-1}(x - \theta_k)$. Then

$$|V_n(\omega, x) - V_n(m_1(x), x)| \leq \sum_{k=0}^{K} \alpha_k \{ |\frac{1}{n_k} \sum_{i=1}^{n_k} (s_{iG}^{(k)}(x) - s_i^{(k)}(x)) D_i^{(k)}(\omega)|$$

$$+ |\frac{1}{n_k} \sum_{i=1}^{n_k} s_i^{(k)} D_i^{(k)}(\omega) - E\{s_i^{(k)} D_i^{(k)}(\omega)\}| \}. \tag{A.1}$$

For any $\delta > 0$,

$$\sup_{d(\omega, m_1(x)) < \delta} \left| \frac{1}{n_k} \sum_{i=1}^{n_k} (s_{iG}^{(k)}(x) - s_i^{(k)}(x)) D_i^{(k)}(\omega) \right| \leq \frac{2\mathrm{diam}(\Omega)\delta}{n_k} \sum_{i=1}^{n_k} \left| W_0^{(k)}(x) + W_1^{(k)}(x)^{\mathrm{T}} X_i^{(k)} \right|,$$

where $W_0^{(k)}(x) := (\overline{X}^{(k)})^{\mathrm{T}} \Sigma_k^{-1}(x - \overline{X}^{(k)}) - \theta_k^{\mathrm{T}} \Sigma_k^{-1}(x - \theta_k)$ and $W_1^{(k)}(x) := \Sigma_k^{-1}(x - \theta_k) - \widehat{\Sigma}_k(x - \overline{X}^{(k)})$. Then

$$s_{iG}^{(k)}(x) - s_i^{(k)}(x) = W_0^{(k)}(x) + (W_1^{(k)}(x))^{\mathrm{T}} X_i^{(k)}. \tag{A.2}$$

We neglect the superscript for the sake of notation simplicity.

Denote $B_{1,M} := \{\|\frac{1}{n}\sum_{i=1}^{n} |X_i|\| \leq M\}$ where $|X_i|$ represents the element-wise absolute value of $X_i$. Let $B_{2,M} = \{\|\widehat{\Sigma}^{-1}\| \leq M\}$, $B_M = B_{1,M} \cap B_{2,M}$ and $\widehat{\Sigma}' = \frac{1}{n}\sum(X_i - \theta)(X_i - \theta)^{\mathrm{T}}$. In $B_{1,M}$,

$$\|\widehat{\Sigma} - \widehat{\Sigma}'\| \leq \|\widehat{\Sigma} - \widehat{\Sigma}'\|_F$$

$$= \left\| \frac{1}{n}\sum_{i=1}^{n} [(X_i - \overline{X})(X_i - \overline{X_i})^{\mathrm{T}} - (X_i - \theta)(X_i - \theta)^{\mathrm{T}}] \right\|_F$$

$$\leq [2p^2\|\overline{X}^2 - \theta^2\|_2^2 + M^2\|\overline{X} - \theta\|_2^2]^{1/2}$$

$$\leq 2p(M + |\theta|)\|\overline{X} - \theta\|_2.$$

Using Theorem 6.5 in [37] leads to

$$P(\|\widehat{\Sigma} - \Sigma\| > \epsilon) \leq 2e^{-C_1\epsilon^2 n} + C_2 e^{-C_3(\frac{1}{2}\epsilon - \sqrt{\frac{C_4}{n}})^2 n},$$

for any $\epsilon \in (2\sqrt{\frac{C_4}{n}}, 1)$. Note that

$$
\begin{aligned}
\|\widehat{\Sigma}^{-1}\| &\leq \|\Sigma^{-1}\| + \|\widehat{\Sigma}^{-1} - \Sigma^{-1}\| \\
&\leq \|\Sigma^{-1}\| + \frac{|\gamma_1(\Sigma) - \gamma_1(\widehat{\Sigma})|}{\gamma_1(\Sigma)\gamma_1(\widehat{\Sigma})} \\
&\leq \|\Sigma^{-1}\| + \frac{\|\Sigma - \widehat{\Sigma}\|}{\gamma_1(\Sigma)(\gamma_1(\Sigma) - \|\Sigma - \widehat{\Sigma}\|)},
\end{aligned}
$$

where the third inequality is from the Weyl's inequality. Then there exists $M > 0$ such that

$$
P(\|\widehat{\Sigma}^{-1}\| \geq M) \leq 2e^{-C_1 \frac{4}{R_3^2} n} + C_2 e^{(-C_3(\frac{2}{R_3} - \sqrt{\frac{C_4}{n}})n)}.
$$

Consequently,

$$
P(B_M^c) \leq 3e^{-\frac{4C_1}{R_3^2}n} + C_2 e^{(-C_3(\frac{2}{R_3} - \sqrt{\frac{C_4}{n}})n)}.
$$

In $B_M$,

$$
\begin{aligned}
W_0(x) &= |(\overline{X} - \theta)\Sigma^{-1}x - \theta\Sigma^{-1}(\theta - \overline{X}) - \overline{X}\Sigma^{-1}(\theta - \overline{X}))| \\
&\leq |(\overline{X} - \theta)\Sigma^{-1}(|x| + |\theta| + M)|.
\end{aligned}
$$

Hence in $B_M' := \cap_{k=0}^K B_M^{(k)}$, we have

$$
P\left(\sum_{k=0}^K \alpha_k |W_0(x)| > t\right) \leq 2e^{-\frac{C_5 t^2}{\Sigma_{k=0}^K \alpha_k^2 \|W_0(x)\|_{\Psi_2}^2}}
$$

$$
\leq 2e^{-\frac{C_6 t^2}{\Sigma_{k=0}^K \frac{n_k}{(n_0 + n_{\mathcal{A}})^2}}} \leq 2e^{-C_6 t^2(n_0 + n_{\mathcal{A}})}, \tag{A.3}
$$

for some constants $C_5, C_6 > 0$.

In the first inequality, we use general Hoeffding's inequality (Theorem 2.6.2 in [35]) and in the second inequality, we use Proposition 2.6.1 in [35].

In addition,

$$
\begin{aligned}
\frac{1}{n}\sum_{i=1}^n |(W_1(x))^T X_i| &\leq M[\|\Sigma^{-1} - \widehat{\Sigma}^{-1}\|\|x\| - \Sigma^{-1}(\theta - \overline{X}) + (\Sigma^{-1} - \widehat{\Sigma}^{-1})\overline{X}] \\
&\leq M(\|x\| + M)(\|\Sigma^{-1} - \widehat{\Sigma}^{-1}\|) + MR_3|\theta - \overline{X}| \\
&\leq M^2(\|x\| + M)R_3\|\Sigma - \widehat{\Sigma}\| + MR_3|\theta - \overline{X}|.
\end{aligned}
$$

Hence in $B_M'$

$$
P\left(\sum_{k=0}^K \alpha_k \frac{1}{n_k}\sum_{i=1}^{n_k} |(X_i^{(k)})^T W_1^{(k)}(x)| > t\right) \leq 2e^{-\frac{C_8 t^2}{n_0 + n_{\mathcal{A}}}} + P\left(\sum_{k=0}^K \alpha_k C_9 \|\Sigma_k - \widehat{\Sigma}_k\| > t\right) \tag{A.4}
$$

for some constant $C_8, C_9 > 0$. Using Markov inequality, the second term is bounded by

$$
e^{-\lambda t}\Pi_{k=0}^K Ee^{\lambda \alpha_k C_9 \|\Sigma_k - \widehat{\Sigma}_k\|} \leq e^{-\lambda t}e^{4pK + \sum_{k=0}^K C_{10}\frac{\alpha_k^2 \lambda^2}{n_k}},
$$

which is from Theorem 6.5 in [37] for any $\lambda$ satisfying $\lambda \leq C_{11}n_*$ for some constants $C_{10}, C_{11} > 0$. We can choose $\lambda := \frac{t(n_0 + n_{\mathcal{A}})}{2C_{10}}$, which does satisfy the theorem's assumption if $t = O\left((n_0 + n_{\mathcal{A}})^{-1/2}\right)$. Here we utilize Condition 2. Then

$$
P\left(\sum_{k=0}^K \alpha_k C_9 \|\Sigma_k - \widehat{\Sigma}_k\| > t\right) \leq e^{-\frac{t^2(n_0 + n_{\mathcal{A}})}{4C_{12}} + 4dK}. \tag{A.5}
$$

Note that $K = o(n_0 + n_\mathcal{A})$, hence

$$\sup_{d(\omega, m_1(x)) < \delta} \sum_{k=0}^{K} \alpha_k \left[ \left\| \frac{1}{n_k} \sum (s_{iG}^{(k)}(x) - s_i^{(k)}(x)) D_i^k(\omega) \right\| \right] = O_p \left( \delta(n_0 + n_\mathcal{A})^{-1/2} \right),$$

since $P(B_M) \to 1$ due to Condition 2.

To bound the second term of (A.1), following the proof of Theorem 2 in [26] for each $k$, we define the functions $g_\omega^{(k)} : \mathbb{R}^p \times \Omega \mapsto \mathbb{R}$ as

$$g_\omega^{(k)}(z, y) = [1 + (z - \theta_k)^\mathrm{T} \Sigma_k^{-1}(x - \theta_k)] d^2(y, \omega)$$

and the function class

$$\mathcal{M}_\delta^{(k)} := \{ g_\omega^{(k)} - g_{m_1(x)}^{(k)} : d(\omega, m_1(x)) < \delta \}.$$

We have

$$E \left\{ \sup_{d(\omega, m_1(x)) < \delta} \left| \frac{1}{n_k} \sum_{i=1}^{n_k} s_{iG}^{(k)}(x) D_{(i)}^{(k)}(g) - E\{ s_i^{(k)} D_i^{(k)}(g) \} \right| \right\} \leq J (E[(G_\delta^{(k)}(x))^2]^{\frac{1}{2}} \sqrt{n_k},$$

where $G_\delta^{(k)}(z) := 2\mathrm{diam}(\Omega)\delta[1 + (z - \theta_k)^\mathrm{T} \Sigma_k^{-1}(x - \theta_k)]$ is the envelop function.

Note that

$$E(G_\delta^{(k)}(X)^2) \leq C_{14}\delta^2$$

for some constant $C_{14} > 0$, which does not depend on $k$. Hence

$$E \left\{ \sup_{d(\omega, m_1(x)) < \delta} \left| \sum_{k=0}^{K} \frac{1}{n_0 + n_\mathcal{A}} \sum_{i=1}^{n_k} s_{iG}^{(k)}(x) D_i^{(k)}(g) - E\{ s_i^{(k)} D_i^{(k)}(g) \} \right| \right\} = O \left( \delta \sum_{k=0}^{K} \frac{\sqrt{n_k}}{n_0 + n_\mathcal{A}} \right). \tag{A.6}$$

Define

$$D_R := \left\{ \sup_{d(\omega, m_1(x)) < \delta} \sum_{k=0}^{K} \alpha_k \left| \frac{1}{n_k} \sum_{i=1}^{n_k} (s_{iG}^{(k)}(x) - s_i^{(k)}(x)) D_i^{(k)}(\omega) \right| \leq R\delta(n_0 + n_\mathcal{A})^{-1/2} \right\}.$$

We have

$$E \left\{ 1_{D_R} \sup_{d(\omega, m_1(x)) < \delta} |V_n(w) - V_n(m_1(x))| \right\} \leq aR\delta \left( \sum_{k=0}^{K} \frac{\sqrt{n_k}}{n_0 + n_\mathcal{A}} + (n_o + n_\mathcal{A})^{-1/2} \right),$$

for some constant $a > 0$.

Next we show

$$|M^1(w, x) - M_n^1(w, x)| = o_p(1),$$

and for all $\omega \in \Omega$ and for any $\kappa > 0, \varphi > 0$, there exists $\delta > 0$ such that

$$\limsup_n P( \sup_{d(\omega_1, \omega_2) < \delta} |M_n^1(\omega_1, x) - M_n^1(\omega_2, x)| > \kappa) \leq \varphi. \tag{A.7}$$

To prove the first assertion, we denote

$$\tilde{M}_n^1(\omega, x) = \frac{1}{n_0 + n_\mathcal{A}} \sum_{k=0}^{K} \sum_{i=1}^{n_k} s_i^{(k)}(x) d^2(\omega, Y_i^{(k)}).$$

Then $E\tilde{M}_n^1(\omega, x) = M_n^1(\omega, x)$ and $\mathrm{Var}(\tilde{M}_n^1(\omega, x)) \leq 2\sum_{k=0}^{K} C' \frac{n_k}{(n_0+n_\mathcal{A})^2}$ for some constant $C' > 0$. Hence $\tilde{M}_n^1(\omega, x) - M_n^1(\omega, x) = o_p(1)$.

Besides,

$$M_n^1(\omega) - \tilde{M}_n^1(\omega) = \sum_{k=0}^{K} \alpha_k \frac{W_0^{(k)}}{n_k} \sum_{i=1}^{n_k} d^2(Y_i^{(k)}, \omega) + \sum_{k=0}^{K} \alpha_k \frac{(W_1^{(k)})^\mathrm{T}}{n_k} \sum_{i=1}^{n_k} X_i^{(k)} d^2(Y_i^{(k)}, \omega)$$

$$= o_p(1).$$

The last equation is true since $\sum_{k=0}^{K} \alpha_k \frac{1}{n_k} \sum_{i=1}^{n_k} |(W_1(x))^{\mathsf{T}} X_i| = o_p(1)$ and $\sum_{k=0}^{K} \alpha_k |W_0(x)| = o_p(1)$.

To prove (A.7), note that for any $\gamma_1, \gamma_2 \in \Omega$,

$$|M_n^1(\gamma_1, x) - M_n^1(\gamma_1, x)| \le 2\mathrm{diam}(\Omega) d(\gamma_1, \gamma_2) \frac{1}{n_0 + n_{\mathcal{A}}} \sum_{k=0}^{K} \sum_{i=1}^{n_k} |s_i^{(k)} + W_0^{(k)} + (W_1^{(k)})^{\mathsf{T}} X_i|$$

$$= O_p(d(\gamma_1, \gamma_2)).$$

The last equation is true since $\sum_{k=0}^{K} \alpha_k \frac{1}{n_k} \sum_{i=1}^{n_k} |(W_1(x))^{\mathsf{T}} X_i| = o_p(1)$ and $\sum_{k=0}^{K} \alpha_k |W_0(x)| = o_p(1)$, and we can prove

$$\sum_{k=0}^{K} \alpha_k \frac{1}{n_k} \sum_{i=1}^{n_k} |s_i^{(k)}| = 1 + o_p(1) \tag{A.8}$$

in a similar way of (A.3). It follows that $d(\widehat{m}_1(x), m_1(x)) = o_p(1)$.

Set $r_n = \left( \sum_{k=0}^{K} \frac{\sqrt{n_k}}{n_0 + n_{\mathcal{A}}} + (n_0 + n_{\mathcal{A}})^{-1/2} \right)^{\frac{\beta}{2(\beta-1)}}$ and

$$S_{j,n}(x) = \{\omega : 2^{j-1} < r_n d(\omega, m_1(x))^{\frac{\beta}{2}} \le 2^j\}.$$

Choose $\eta > 0$ to satisfy (iii) in Condition 4 and also small enough that (ii) in Condition 4 holds for all $\delta < \eta$ and set $\tilde{\eta} := \eta^{\frac{\beta}{2}}$. For any integer $L_2$,

$$P\left( r_n d^{\beta/2}(\widehat{m}_1(x), m_1(x)) > 2^{L_2} \right) \le P(D_R^c) + P\left( 2d(\widehat{m}_1(x), m_1(x)) \ge \eta \right)$$

$$+ \sum_{\substack{j \ge L_2 \\ 2^j \le r_n \tilde{\eta}}} P\left( \{ \sup_{\omega \in S_{j,n}} |V_n(\omega) - V_n(m_1(x))| \ge C \frac{2^{2(j-1)}}{r_n^2} \} \cap D_R \right).$$

The second term converges to 0 since $d(\widehat{m}_1(x), m_1(x)) = o_p(1)$. For each $j$ in the sum in the third term, we have $d(\omega, m_1(x)) \le (\frac{2^j}{r_n})^{\frac{2}{\beta}} \le \eta$, so the sum is bounded by

$$4aC^{-1} \sum_{\substack{j \ge L_2 \\ 2^j \le r_n \tilde{\eta}}} \frac{2^{\frac{2j(1-\beta)}{\beta}}}{r_n^{\frac{2(1-\beta)}{\beta}} \left( \sum_{k=0}^{K} \frac{\sqrt{n_k}}{n_0 + n_{\mathcal{A}}} + (n_o + n_{\mathcal{A}})^{-\frac{1}{2}} \right)} \le 4aC^{-1} \sum_{j \ge L_2} \left( \frac{1}{4^{\frac{\beta-1}{\beta}}} \right)^j.$$

Choose $L_2$ large enough, this probability can be small enough.

Hence we have

$$d^2(\widehat{m}_1(x), m_1(x)) = O_p\left( \frac{\sum_{k=0}^{K} \sqrt{n_k}}{n_0 + n_{\mathcal{A}}} + (n_0 + n_{\mathcal{A}})^{-1/2})^{\frac{1}{\beta-1}} \right).$$

$\square$

### C.3.2  Proof of Theorem 1 Given Lemma 3

*Proof.* The proof is similar to that of Lemma 1. First, it is easy to check that $f \in BV((0,1), H_2)$ for some large $H_2 > 0$ since

$$f = E\left\{ \sum_{k=0}^{K} \alpha_k s_G^{(k)}(x) \mathbf{1}_{\{\sum_{k=0}^{K} \alpha_k s_G^{(k)}(x) > 0\}} F_{\nu^{(0)}}^{-1} \right\} - E\left\{ \left| \sum_{k=0}^{K} \alpha_k s_G^{(k)}(x) \right| \mathbf{1}_{\{\sum_{k=0}^{K} \alpha_k s_G^{(k)}(x) < 0\}} F_{\nu^{(0)}}^{-1} \right\}$$

and $E|\sum_{k=0}^{K} \alpha_k s_G^{(k)}(x)| < \infty$. Taking $H_2$ large enough, we also claim that $P(\widehat{f} \in BV((0,1), H_2)) \to 1$ since $\widehat{f}_0$ also has a similar decomposition and

$$\frac{1}{n_0 + n_{\mathcal{A}}} \sum_{k=0}^{K} \sum_{i=1}^{n_k} |s_{iG}^{(k)}(x)| \le \frac{1}{n_0 + n_{\mathcal{A}}} \sum_{i=1}^{n_k} (|s_{iG}^{(k)}(x) - s_i^{(k)}(x)| + |s_i^{(k)}(x)|)$$

$$\le o_p(1) + H_3,$$

for some $H_3 > 0$ with high probability. The last inequality follows from (A.2), (A.8), (A.5) and (A.4). Thus, (i) in Condition 4 holds.

(ii) in Condition 4 follows from the same arguments used in the proof of Lemma 1. It remains to verify (iii) in Condition 4, which holds since

$$M^1(g, x) - M^1(f, x) = \|f - g\|_2^2.$$

Hence the result follows by using Lemma 3. $\qquad\square$

## C.4  Proof of Theorem 2

*Proof.* Recall that $f^{(0)}(x) = E\{s_G^{(0)}(x)F_{\nu^{(0)}}^{-1}\}$. Using the definition of $\widehat{f}_0$,

$$
\begin{aligned}
\|\widehat{f}_0 - f^{(0)}\|_2^2 &= \frac{1}{n_0}\sum_{i=1}^{n_0} s_{iG}^{(0)}(x)(\|\widehat{f}_0 - F_{\nu_i^{(0)}}^{-1}\|_2^2 - \|f^{(0)} - F_{\nu_i^{(0)}}^{-1}\|_2^2) \\
&\quad + \frac{2}{n_0}\sum_{i=1}^{n_0} s_{iG}^{(0)}(x)\langle F_{\nu_i^{(0)}}^{-1} - f^{(0)}, \widehat{f}_0 - f^{(0)}\rangle_2 \\
&\leq -\lambda\|\widehat{f}_0 - \widehat{f}\|_2 + \lambda\|f^{(0)} - \widehat{f}\|_2 \\
&\quad + \frac{2}{n_0}\sum_{i=1}^{n_0} s_{iG}^{(0)}(x)\langle F_{\nu_i^{(0)}}^{-1} - f^{(0)}, \widehat{f}_0 - f^{(0)}\rangle_2 \\
&\leq -\lambda\|\widehat{f}_0 - \widehat{f}\|_2 + \lambda\|f^{(0)} - \widehat{f}\|_2 \\
&\quad + 2\|\widehat{f}_0 - f^{(0)}\|_2 \| \frac{1}{n_0}\sum_{i=1}^{n_0} s_{iG}^{(0)}(x)F_{\nu_i^{(0)}}^{-1} - f^{(0)}\|_2.
\end{aligned}
$$

We define the event $E_n := \{\|n_0^{-1}\sum_{i=0}^{n_0} s_{iG}^{(0)}(x)F_{\nu_i^0}^{-1} - f^{(0)}\|_2 \leq \lambda/2\}$. According to Lemma 1, we have $P(E_n) \to 1$ since $\lambda \asymp n_0^{-1/2+\epsilon}$.

Under $E_n$ for $n$ large enough,

$$\|\widehat{f}_0 - f^{(0)}\|_2^2 \leq 2\lambda\|f^{(0)} - \widehat{f}\|_2 \leq 2\lambda(\|f^{(0)} - f\|_2 + \|\widehat{f} - f\|_2)$$

holds. Hence we have

$$d_{\mathcal{W}}^2(\widehat{m}_G^{(0)}(x), m_G^{(0)}(x)) \leq \|\widehat{f}_0 - f^{(0)}\|_2^2 = O_p\Big(n_0^{-1/2+\epsilon}(\|\widehat{f} - f\|_2 + \psi)\Big).$$

Then using Theorem 1, it follows that

$$d_{\mathcal{W}}^2(\widehat{m}_G^{(0)}(x), m_G^{(0)}(x)) = O_p\Big(n_0^{-1/2+\epsilon}(\psi + \frac{\sum_{k=0}^K \sqrt{n_k}}{n_0 + n_{\mathcal{A}}} + (n_0 + n_{\mathcal{A}})^{-1/2})\Big).$$

$\qquad\square$

## C.5  Proof of Theorem 3

*Proof.* We first prove $P(\widehat{\mathcal{A}} = \mathcal{A}) \to 1$. Note that

$$
\begin{aligned}
P(\widehat{\mathcal{A}} = \mathcal{A}) &\geq P(\max_{k\in\mathcal{A}} \widehat{\psi}_k < \min_{k\in\mathcal{A}^C} \widehat{\psi}_k) \\
&\geq 1 - P(\max_{1\leq k\leq K} |\widehat{\psi}_k - \psi_k| > |\max_{k\in\mathcal{A}} \psi_k - \min_{k\in\mathcal{A}^C} \psi_k|) \\
&\geq 1 - K\max_{1\leq k'\leq K} P(|\widehat{\psi}_{k'} - \psi_{k'}| > |\max_{k\in\mathcal{A}} \psi_k - \min_{k\in\mathcal{A}^C} \psi_k|).
\end{aligned}
$$

According to Lemma 1, $|\psi_k - \widehat{\psi}_k| = O_p(n_k^{-1/2} + n_0^{-1/2})$. According to Condition 3,

$$\frac{|\max_{k\in\mathcal{A}} \psi_k - \min_{k\in\mathcal{A}^C} \psi_k|}{n_*^{-1/2} + n_0^{-1/2}} \to \infty.$$

Hence
$$P(\widehat{\mathcal{A}} = \mathcal{A}) = 1 - o(1).$$
Then using Theorem 2 where we substitute $\mathcal{A}$ for $\{1, \ldots, K\}$, we have

$$d_{\mathcal{W}}^2(\widehat{m}_G^{(0)}(x), m_G^{(0)}(x)) = O_p\Big(n_0^{-1/2+\epsilon}\big(\max_{k \in \mathcal{A}} \psi_k + \frac{\sum_{k \in \mathcal{A} \cup \{0\}} \sqrt{n_k}}{\sum_{k \in \mathcal{A} \cup \{0\}} n_k} + \big(\sum_{k \in \mathcal{A} \cup \{0\}} n_k\big)^{-1/2}\big)\Big).$$

$\square$

# D   Transfer Learning for Local Fréchet Regression

For simplicity, we consider a scalar predictor $X^{(k)} \in \mathbb{R}$ for $k = 0, \ldots, K$. For predictors in $\mathbb{R}^p$, the explicit form of the weight function can be found in Section 2.3 of [17]. Under the same source-target data setting, we define

$$m^{(k)}(x) = \arg\min_{\mu \in \mathcal{W}} E\{d_{\mathcal{W}}^2(\nu^{(k)}, \mu) | X^{(k)} = x\},$$

$$m_{L,h}^{(k)}(x) = \arg\min_{\mu \in \mathcal{W}} E\{s_L^{(k)}(x, h) d_{\mathcal{W}}^2(\nu^{(k)}, \mu)\},$$

$$\widehat{m}_{L,h}^{(k)}(x) = \arg\min_{\mu \in \mathcal{W}} \frac{1}{n_k} \sum_{i=1}^{n_k} s_{iL}^{(k)}(x, h) d_{\mathcal{W}}^2(\nu_i^{(k)}, \mu)\},$$

where $s_L^{(k)}(x)$ and $s_{iL}^{(k)}(x)$ represents the population and sample weight functions of local Fréchet regression for the $k$th source.

For notational simplicity, define

$$f_{\oplus}^{(k)}(x) = E\{F_{\nu^{(k)}}^{-1} | X^{(k)} = x\},$$

$$f_h^{(k)}(x) = E\{s_L^{(k)}(x, h) F_{\nu^{(k)}}^{-1}\},$$

$$f_h(x) = \sum_{k=0}^{K} \alpha_k f_h^{(k)}(x).$$

Similar to Section 3, we introduce the Local Wasserstein Transfer Learning (LWaTL) algorithm in Algorithm 3, assuming that all sources are informative. When the informative set is unknown, we incorporate an additional step to identify the informative set, as outlined in Algorithm 4. Theoretical guarantees for these algorithms are provided in Theorem 4 and Theorem 5.

We impose the following condition, where the first two parts correspond to the kernel and distributional assumptions that are standard in local linear regression.

**Condition 5.**   *(i) The kernel $K$ is a probability density function, symmetric around zero. Furthermore, defining $K_{sj} = \int_{\mathbb{R}} K^s(u) u^j du$, $|K_{14}|$ and $|K_{26}|$ are both finite.*

*(ii) The marginal density $q^{(k)}$ of $X^{(k)}$ and the conditional density of $X^{(k)}$ given $\nu^{(k)}$, $g_{\nu^{(k)}}^{(k)}$, exist and are twice continuously differentiable. Besides, it satisfies that*

$$\sup_{0 \le k \le K} \sup_{z, \nu^{(k)}} \max\{(q^{(k)})''(z), (g_{\nu^{(k)}}^{(k)})''(z)\} < H_3$$

*for some $H_3 > 0$.*

*(iii) $h \to 0$ and $n_0 h \to \infty$.*

*(iv) $\frac{\sum_{k=1}^{K} n_k^{-1}}{h} = o(1)$.*

*(v) $\nu^{(k)}$ have a common bounded domain.*

*(vi) $u_j^{(k)} := E\big(K_h(X_i^{(k)} - x)(X_i^{(k)} - x)^j\big)$, $j = 0, 1, 2$ and $\sigma_0^{(k)} := u_0^{(k)} u_2^{(k)} - (u_1^{(k)})^2 > 0$ are bounded.*

**Algorithm 3** Local Wasserstein Transfer Learning (LWaTL)

---

**Input:** Target and source data $\{(x_i^{(0)}, \nu_i^{(0)})\}_{i=1}^{n_0} \cup \left( \cup_{1 \leq k \leq K} \{(x_i^{(k)}, \nu_i^{(k)})\}_{i=1}^{n_k} \right)$, regularization parameter $\lambda$, bandwidth $h$ and query point $x \in \mathbb{R}$.

**Output:** Target estimator $\widehat{m}_{L,h}^{(0)}(x)$.

1: Weighted auxiliary estimator

$$\widehat{f}_h(x) = \frac{1}{n_0 + n_{\mathcal{A}}} \sum_{k=0}^{K} n_k \widehat{f}_h^{(k)}(x),$$

where $\widehat{f}_h^{(k)}(x) = n_k^{-1} \sum_{i=1}^{n_k} s_{iL}^{(k)}(x, h) F_{\nu_i^{(k)}}^{-1}$ and $n_{\mathcal{A}} = \sum_{k=1}^{K} n_k$.

2: Bias correction using target data

$$\widehat{f}_{0h}(x) = \operatorname*{arg\,min}_{g \in L^2(0,1)} \frac{1}{n_0} \sum_{i=1}^{n_0} s_{iL}^{(0)}(x, h) \| F_{\nu_i^{(0)}}^{-1} - g \|_2^2 + \lambda \| g - \widehat{f}_h(x) \|_2.$$

3: Projection to Wasserstein space

$$\widehat{m}_{L,h}^{(0)}(x) = \operatorname*{arg\,min}_{\mu \in \mathcal{W}} \left\| F_\mu^{-1} - \widehat{f}_{0h}(x) \right\|_2.$$

---

The following theorem establishes the rate of convergence for the LWaTL algorithm.

**Theorem 4.** *Assume Condition 5 holds and the regularization parameter satisfies $\lambda \asymp n_0^{-1/2} h^{-1/2-\epsilon}$ for some $\epsilon > 0$. Then, for the LWaTL algorithm and a fixed $x \in \mathbb{R}^p$, we have*

$$d_{\mathcal{W}}^2(\widehat{m}_{L,h}^{(0)}(x), m^{(0)}(x)) = O_p\left( h^4 + n_0^{-1/2} h^{-1/2-\epsilon} \left[ \psi_L + h^2 + h^{-1/2} \left( \left( \sum_{k=0}^{K} n_k^{-1} \right)^{1/2} + \sum_{k=0}^{K} \frac{\sqrt{n_k}}{n_0 + n_{\mathcal{A}}} \right) \right] \right),$$

*where $\psi_L = \max_{1 \leq k \leq K} \| f_\oplus^{(0)}(x) - f_\oplus^{(k)}(x) \|$.*

*Proof.* Note that

$$d_{\mathcal{W}}^2(\widehat{m}_{L,h}^{(0)}(x), m^{(0)}(x)) \leq 2\{ d_{\mathcal{W}}^2(m_{L,h}^{(0)}(x), m^{(0)}(x)) + d_{\mathcal{W}}^2(m_{L,h}^{(0)}(x), \widehat{m}_{L,h}^{(0)}(x)) \}$$

$$\leq 2\{ d_{\mathcal{W}}^2(m_{L,h}^{(0)}(x), m^{(0)}(x)) + \| f_h^{(0)}(x) - \widehat{f}_{0h}(x) \|_2^2 \}.$$

For the first term, We could use Theorem 3 of [26] since we can check all its assumptions easily, which is similar to the proof of Lemma 1. Here we should consider the metric space $BV((0, 1), H_4)$ for some $H_4 > 0$ endowed with the probability measure naturally induced by the canonical embedding of the Wasserstein space.

For the second term, first note that using Theorem 4 of [26], we have $\| f_h^{(0)}(x) - \widehat{f}_h^{(0)}(x) \|_2 = O_p\left( (n_0 h)^{-1/2} \right)$. Then similar to the proof of Theorem 2, we have

$$\| f_h^{(0)}(x) - \widehat{f}_{0h}(x) \|_2^2 \leq -\lambda \| \widehat{f}_{0h}(x) - \widehat{f}_h(x) \|_2 + \lambda \| f_h^{(0)}(x) - \widehat{f}_h(x) \|_2$$

$$+ 2\| \widehat{f}_h^{(0)}(x) - f_h^{(0)}(x) \|_2 \| \widehat{f}_{0h}(x) - f_h^{(0)}(x) \|_2.$$

Hence in $E_n := \{ \| f_h^{(0)}(x) - \widehat{f}_h^{(0)}(x) \|_2 \leq \frac{1}{2}\lambda \}$,

$$\| f_h^{(0)}(x) - \widehat{f}_{0h}(x) \|_2^2 \leq 2\lambda \| f_h^{(0)}(x) - \widehat{f}_h(x) \|_2 \leq 2\lambda (\psi_h + \| f_h(x) - \widehat{f}_h(x) \|_2), \tag{A.9}$$

where $\psi_h = \max_{1 \leq k \leq K} \| f_h^{(k)} - f_h^{(0)} \|_2$. For the first term of (A.9), using Theorem 3 of [26] and (i), (ii) in Condition 5, we have

$$\psi_h = \psi_L + O(h^2).$$

For the second term of (A.9), we could follow the proof of Lemma 3 to study the asymptotic rate of convergence. The difference is that we do not assume covariates are sub-Gaussian but

**Algorithm 4** Adaptive Local Wasserstein Transfer Learning (ALWaTL)

---

**Input:** Target and source data $\{(x_i^{(0)}, \nu_i^{(0)})\}_{i=1}^{n_0} \cup \left( \cup_{1 \leq k \leq K} \{(x_i^{(k)}, \nu_i^{(k)})\}_{i=1}^{n_k} \right)$, regularization parameter $\lambda$, bandwidth $h$, number of informative sources $L$, and query point $x \in \mathbb{R}$.

**Output:** Target estimator $\widehat{m}_{L,h}^{(0)}(x)$.

1: Compute discrepancy scores. For each source dataset $k = 1, \ldots, K$, compute the empirical discrepancy
$$\widehat{\psi}_{k,h} = \|\widehat{f}_h^{(0)}(x) - \widehat{f}_h^{(k)}(x)\|_2,$$
where $\widehat{f}_h^{(k)}(x) = n_k^{-1} \sum_{i=1}^{n_k} s_{iL}^{(k)}(x,h) F_{\nu_i^{(k)}}^{-1}$. Construct the adaptive informative set by selecting the $L$ smallest discrepancy scores
$$\widehat{\mathcal{A}} = \{1 \leq k \leq K : \widehat{\psi}_{k,h} \text{ is among the smallest } L \text{ values}\}.$$

2: Weighted auxiliary estimator
$$\widehat{f}_h(x) = \frac{1}{\sum_{k \in \widehat{\mathcal{A}} \cup \{0\}} n_k} \sum_{k \in \widehat{\mathcal{A}} \cup \{0\}} n_k \widehat{f}_h^{(k)}(x).$$

3: Bias correction using target data
$$\widehat{f}_{0h}(x) = \operatorname*{arg\,min}_{g \in L^2(0,1)} \frac{1}{n_0} \sum_{i=1}^{n_0} s_{iL}^{(0)}(x,h)\|F_{\nu_i^{(0)}}^{-1} - g\|_2^2 + \lambda\|g - \widehat{f}_h(x)\|_2.$$

4: Projection to Wasserstein space
$$\widehat{m}_{L,h}^{(0)}(x) = \operatorname*{arg\,min}_{\mu \in \mathcal{W}} \left\| F_\mu^{-1} - \widehat{f}_{0h}(x) \right\|_2.$$

---

we have upper bounds for moments related to covariates and the kernel. In detail, we define $\tilde{s}_{iL}^{(k)}(x,h) := K_h(X_i^{(k)} - x)\frac{u_0^{(k)} - u_1^{(k)}(X_i^{(k)} - x)}{(\sigma_0^{(k)})^2}$ and $D_i^{(k)}(g,x) := \|F_{\nu_i^{(k)}}^{-1} - g\|_2^2 - \|F_{\nu_i^{(k)}}^{-1} - f_h^{(k)}(x)\|_2^2$. Then similar to (A.6) and the proof of Theorem 4 in [26] we have for small $\delta > 0$,

$$E\left\{ \sup_{d_{L^2}(g, f_h^{(k)}) < \delta} \left| \frac{1}{n_0 + n_\mathcal{A}} \sum_{k=0}^{K} \sum_{i=1}^{n_k} \tilde{s}_{iL}^{(k)}(x,h) D_i^{(k)}(g,x) - \sum_{k=0}^{K} \alpha_k E\{\tilde{s}_{iL}^{(k)}(x,h) D_i^{(k)}(g,x)\} \right| \right\}$$
$$= O\left( \delta h^{-1/2} \frac{\sum_{k=0}^{K} \sqrt{n_k}}{n_0 + n_\mathcal{A}} \right).$$

Another term we need to bound is $\sum_{k=0}^{K} \sum_{i=1}^{n_k} |s_{iL}^{(k)}(x,h) - \tilde{s}_{iL}^{(k)}(x,h)|$. Note that we can define $B_M := \cap_{k=0}^{K}(\{|(\widehat{\sigma}_0^{(k)})^2| > \frac{1}{2}(\sigma_0^{(k)})^2\} \cap_{j=0}^{2} (\{\frac{1}{n_k} \sum_{i=1}^{n_k} |K_h(X_i^{(k)} - x)(X_i^{(k)} - x)^j| < h^j M\}))$ for some large $M > 0$. Then it is easy to check that
$$P\left((B_M)^c\right) \leq \sum_{k=0}^{K} \frac{C_{A1}}{n_k},$$
for some $C_{A1} > 0$. In $B_M$, observe that
$$|s_{iL}^{(k)}(x,h) - \tilde{s}_{iL}^{(k)}(x,h)| \leq |W_{0n}^{(k)} K_h(X_i^{(k)} - x)| + |W_{1n}^{(k)} K_h(X_i^{(k)} - x)(X_i^{(k)} - x)|,$$
where
$$W_{0n}^{(k)} := \frac{\widehat{u}_2^{(k)}}{(\widehat{\sigma}_0^{(k)})^2} - \frac{u_2^{(k)}}{(\sigma_0^{(k)})^2}, \quad W_{1n}^{(k)} := \frac{\widehat{u}_1^{(k)}}{(\widehat{\sigma}_0^{(k)})^2} - \frac{u_1^{(k)}}{(\sigma_0^{(k)})^2}.$$
Note that in $B_M$, it is easy to check
$$|W_{0n}^{(k)}| \leq C_{A2}\left( \sum_{j=0}^{2} h^{-j} \left| u_j^{(k)} - \widehat{u}_j^{(k)} \right| \right)$$

and

$$|W_{1n}^{(k)}| \le C_{A2}\Big(\sum_{j=0}^{2} h^{-1-j}\Big|u_j^{(k)} - \widehat{u}_j^{(k)}\Big|\Big)$$

for some constants $C_{A2} > 0$. Hence in $B_M$,

$$P\Big(\sum_{k=0}^{K}\sum_{i=1}^{n_k} |s_{iL}^{(k)}(x,h) - \tilde{s}_{iL}^{(k)}(x,h)| > t\Big)$$

$$\le P\Big(C_{A3}\sum_{k=0}^{K}\alpha_k(|W_{0n}| + h|W_{1n}|) > t\Big)$$

$$\le \frac{1}{t^2 h}C_{A4}\sum_{k=0}^{K}\frac{1}{n_k}.$$

Hence $\sum_{k=0}^{K}\sum_{i=1}^{n_k} |s_{iL}^{(k)}(x,h) - \tilde{s}_{iL}^{(k)}(x,h)| = O_p(h^{-1/2}\sum_{k=0}^{K} n_k^{-1})$. In addition, we can check that $\widehat{f}_h(x) \in BV((0,1), H_5)$ exists almost surely for some $H_5 > 0$ by showing $\frac{1}{n}\sum_{k=1}^{K}\sum_{i=1}^{n_k} |s_{iL}^{(k)}(x,h)| = O_p(1)$. Also utilizing the same technique in the proof of Lemma 3, It suffices to show that $\|f_h(x) - \widehat{f}_h(x)\|_2 = o_p(1)$. It is just a simple generalization of Lemma 2 in [26]. Eventually, we have

$$\|f_h^{(0)}(x) - \widehat{f}_{0h}(x)\|_2 = O_p\left(h^{-1/2}\Big((\sum_{k=0}^{K} n_k^{-1})^{1/2} + \sum_{k=0}^{K}\frac{\sqrt{n_k}}{n_0 + n_{\mathcal{A}}}\Big)\right)$$

and

$$d_{\mathcal{W}}^2(\widehat{m}_{L,h}^{(0)}(x), m^{(0)}(x)) = O_p\left(h^4 + n_0^{-1/2}h^{-1/2-\epsilon}\Big[\psi_L + h^2 + h^{-1/2}\Big((\sum_{k=0}^{K} n_k^{-1})^{1/2} + \sum_{k=0}^{K}\frac{\sqrt{n_k}}{n_0 + n_{\mathcal{A}}}\Big)\Big]\right).$$

$\square$

*Remark* 3. Theorem 4 can be extended to general metric spaces and serves as a parallel version of Lemma 3. Besides, if $n_*$ is much larger than $n_0$ and $\psi_L = O(n_0^{-k})$ with $k > \frac{6}{15-10\epsilon}$, then among bandwidth sequences $h = n_0^{-r}$, the optimal sequence is achieved at $r^* = \frac{k}{2}$, leading to the convergence rate

$$d_{\mathcal{W}}^2(\widehat{m}_{L,h}^{(0)}(x), m^{(0)}(x)) = O_p(n_0^{-\frac{3k-2\epsilon k+2}{4}}),$$

which surpasses the convergence rate of local Fréchet regression $O_p(n_0^{-4/5})$ [26].

There is also a parallel version of Theorem 3, built upon the following condition.

**Condition 6.** *Suppose the regularization parameter satisfies* $\lambda \asymp n_0^{-1/2}h^{-1/2-\epsilon}$ *for some* $\epsilon > 0$ *and for some* $\epsilon' > \epsilon$, *There exists a non-empty subset* $\mathcal{A} \subset \{1, \ldots, K\}$ *such that*

$$\frac{h^2 + (n_*h)^{-1/2} + (n_0h)^{-1/2}}{\min_{k\in\mathcal{A}^C}\psi_{k,L}} = o(1), \quad \frac{\max_{k\in\mathcal{A}}\psi_{k,L}}{h^2 + (n_*h)^{-1/2} + n_0^{-1/2}h^{-1/2+\epsilon'}} = O(1),$$

*where* $\mathcal{A}^C = \{1, \ldots, K\}\backslash\mathcal{A}$, $n_* = \min_{1\le k\le K} n_k$ *and* $\psi_{k,L} = \|f_{\oplus}^{(0)}(x) - f_{\oplus}^{(k)}(x)\|_2$.

**Theorem 5.** *Assume Conditions 5 and 6 hold. Then for the ALWaTL algorithm we have*

$$d_{\mathcal{W}}^2(\widehat{m}_{L,h}^{(0)}(x), m(x))$$

$$= O_p\left(h^4 + n_0^{-1/2}h^{-1/2-\epsilon}\Big[\max_{k\in\mathcal{A}}\psi_{k,L} + h^2\right.$$

$$\left. + h^{-1/2}\Big((\sum_{k\in\mathcal{A}\cup\{0\}} n_k^{-1})^{1/2} + \sum_{k\in\mathcal{A}\cup\{0\}}\frac{\sqrt{n_k}}{\sum_{k\in\mathcal{A}\cup\{0\}} n_k}\Big)\Big]\right).$$

*Proof.* We first prove $P(\widehat{\mathcal{A}} = \mathcal{A}) \to 1$. Note that

$$
\begin{aligned}
P(\widehat{\mathcal{A}} = \mathcal{A}) &\geq P(\max_{k \in \mathcal{A}} \widehat{\psi}_{k,h} < \min_{k \in \mathcal{A}^C} \widehat{\psi}_{k,h}) \\
&\geq 1 - P(\max_{1 \leq k \leq K}(|\widehat{\psi}_{k,h} - \psi_{k,h}| + |\psi_{k,h} - \psi_{k,L}|) > |\max_{k \in \mathcal{A}} \psi_{k,L} - \min_{k \in \mathcal{A}^C} \psi_{k,L}|) \\
&\geq 1 - K \max_{1 \leq k' \leq K} P(|\widehat{\psi}_{k'} - \psi_{k'}| + |\psi_{k',h} - \psi_{k',L}|) > |\max_{k \in \mathcal{A}} \psi_{k,L} - \min_{k \in \mathcal{A}^C} \psi_{k,L}|).
\end{aligned}
$$

According to Theorem 3 of [26], $|\psi_{k,h} - \psi_{k,L}| = O(h^2)$. Utilizing Theorem 4 of [26], $|\psi_{k,h} - \widehat{\psi}_{k,h}| = O_p((n_k h)^{-1/2} + (n_0 h)^{-1/2})$. According to Condition 6,

$$
\frac{|\max_{k \in \mathcal{A}} \psi_{k,L} - \min_{k \in \mathcal{A}^C} \psi_{k,L}|}{h^2 + (n_* h)^{-1/2} + (n_0 h)^{-1/2}} \to \infty.
$$

Hence

$$
P(\widehat{\mathcal{A}} = \mathcal{A}) = 1 - o(1).
$$

Then utilizing Theorem 4 where we substitute $\mathcal{A}$ for $\{1, \ldots, K\}$, we have

$$
\begin{aligned}
&d_{\mathcal{W}}^2(\widehat{m}_{L,h}^{(0)}(x), m^{(0)}(x)) \\
&= O_p\left(h^4 + n_0^{-1/2} h^{-1/2-\epsilon}\left[\max_{k \in \mathcal{A}} \psi_{k,L} + h^2\right.\right. \\
&\left.\left.+ h^{-1/2}\left(\left(\sum_{k \in \mathcal{A} \cup \{0\}} n_k^{-1}\right)^{1/2} + \sum_{k \in \mathcal{A} \cup \{0\}} \frac{\sqrt{n_k}}{\sum_{k \in \mathcal{A} \cup \{0\}} n_k}\right)\right]\right).
\end{aligned}
$$

$\square$

*Remark* 4. If $n_*$ is much larger than $n_0$, then the simplified rate is $O_p(h^4 + n_0^{-1/2} h^{3/2-\epsilon} + n_0^{-1} h^{-1+\epsilon'-\epsilon})$. Among bandwidth sequences $h = n_0^r$, the optimal choice is achieved at $r^* = -\frac{1}{5-2\epsilon'}$, leading to the convergence rate

$$
d_{\mathcal{W}}^2(\widehat{m}_{L,h}^{(0)}(x), m(x)) = O_p(n_0^{-\frac{4-\epsilon-\epsilon'}{5-2\epsilon'}}),
$$

which is faster than the convergence rate of local Fréchet regression $O_p(n_0^{-4/5})$ [26].

# E   Transfer Learning for Wasserstein Regression with Empirical Measures

Fréchet regression assumes that each distribution is fully observed. However, this assumption is often impractical, since in real-world settings one rarely encounters datasets where each observation itself constitutes a full distribution. To address, Wasserstein regression with empirical distributions has been proposed [43], where distributions are replaced by their empirical versions, reaching a convergence rate of $O_p(n^{-1/2} + N_{\min}^{-1/4})$, where $N_{\min}$ representing the minimum number of observations. In this section we do the same modification to handle this obstacle, providing algorithms for global and local Wasserstein transfer learning with empirical measures (WaTL-EM), respectively. The only difference between them and those with full distributions is that we substitute empirical distributions $\widehat{\nu}_i^{(k)}$ defined as $\frac{1}{N_i} \sum_{i=1}^{N_i^{(k)}} \delta_{y_{ij}^{(k)}}$ for the true but unobservable distributions $\nu_i^{(k)}$, $1 \leq i \leq n_k, 1 \leq k \leq K$, where we recall the settings in Subsection 3.2 and Appendix D, additionally assume $\{y_{ij}^{(k)}\}_{j=1}^{N_i^{(k)}}$ are independently sampled from $\nu_i^{(k)}$ and define $\delta_z$ as the Dirac measure at $z$. In addition, we denote $N_{\min}$ as $\min_{i,k} N_i^{(k)}$.

**Theorem 6.** *Assume Conditions 1 and 2 hold and the regularization parameter satisfies $\lambda \asymp (n_0^{-1/2} + N_{\min}^{-1/4})^{1-\epsilon}$ for some $\epsilon > 0$. Then, for the WaTL-EM algorithm and a fixed $x \in \mathbb{R}^p$, it holds that*

$$
d_{\mathcal{W}}^2(\widehat{m}_{G,EM}^{(0)}(x), m_G^{(0)}(x)) = O_p\left((n_0^{-1/2} + N_{\min}^{-1/4})^{1-\epsilon}\left(\psi + \frac{\sum_{k=0}^K \sqrt{n_k}}{n_0 + n_{\mathcal{A}}} + (n_0 + n_{\mathcal{A}})^{-1/2} + N_{\min}^{-1/4}\right)\right).
$$

**Algorithm 5** Wasserstein Transfer Learning with Empirical measures (WaTL-EM)

**Input:** Target and source data $\{(x_i^{(0)}, \{y_{ij}^{(0)}\}_{j=1}^{N_i^{(0)}})\}_{i=1}^{n_0} \cup \left( \cup_{1 \leq k \leq K} \{(x_i^{(k)}, \{y_{ij}^{(k)}\}_{j=1}^{N_i^{(k)}}\}_{i=1}^{n_k}\right)$, regularization parameter $\lambda$, and query point $x \in \mathbb{R}^p$.

**Output:** Target estimator $\widehat{m}_{G,EM}^{(0)}(x)$.

1: Empirical measures: $\widehat{\nu}_i^{(k)} = \frac{1}{N_i^{(k)}} \sum_{j=1}^{N_i^{(k)}} \delta_{y_{ij}^{(k)}}$.

2: Weighted auxiliary estimator: $\widehat{f}_{EM}(x) = \frac{1}{n_0+n_{\mathcal{A}}} \sum_{k=0}^{K} n_k \widehat{f}_{EM}^{(k)}(x)$, where $\widehat{f}_{EM}^{(k)}(x) = n_k^{-1} \sum_{i=1}^{n_k} s_{iG}^{(k)}(x) F_{\widehat{\nu}_i^{(k)}}^{-1}$.

3: Bias correction using target data: $\widehat{f}_{0,EM}(x) = \arg\min_{g \in L^2(0,1)} \frac{1}{n_0} \sum_{i=1}^{n_0} s_{iG}^{(0)}(x) \| F_{\widehat{\nu}_i^{(0)}}^{-1} - g \|_2^2 + \lambda \| g - \widehat{f}_{EM}(x) \|_2$.

4: Projection to Wasserstein space: $\widehat{m}_{G,EM}^{(0)}(x) = \arg\min_{\mu \in \mathcal{W}} \left\| F_{\mu}^{-1} - \widehat{f}_{0,EM}(x) \right\|_2$.

---

**Algorithm 6** Local Wasserstein Transfer Learning with Empirical measures (LWaTL-EM)

**Input:** Target and source data $\{(x_i^{(0)}, \{y_{ij}^{(0)}\}_{j=1}^{N_i^{(0)}})\}_{i=1}^{n_0} \cup \left( \cup_{1 \leq k \leq K} \{(x_i^{(k)}, \{y_{ij}^{(k)}\}_{j=1}^{N_i^{(k)}}\}_{i=1}^{n_k}\right)$, regularization parameter $\lambda$, bandwidth $h$ and query point $x \in \mathbb{R}$.

**Output:** Target estimator $\widehat{m}_{L,h}^{(0)}(x)$.

1: Empirical measures: $\widehat{\nu}_i^{(k)} = \frac{1}{N_i^{(k)}} \sum_{j=1}^{N_i^{(k)}} \delta_{y_{ij}^{(k)}}$.

2: Weighted auxiliary estimator

$$\widehat{f}_{h,EM}(x) = \frac{1}{n_0 + n_{\mathcal{A}}} \sum_{k=0}^{K} n_k \widehat{f}_{h,EM}^{(k)}(x),$$

where $\widehat{f}_{h,EM}^{(k)}(x) = n_k^{-1} \sum_{i=1}^{n_k} s_{iL}^{(k)}(x,h) F_{\widehat{\nu}_i^{(k)}}^{-1}$ and $n_{\mathcal{A}} = \sum_{k=1}^{K} n_k$.

3: Bias correction using target data

$$\widehat{f}_{0h,EM}(x) = \arg\min_{g \in L^2(0,1)} \frac{1}{n_0} \sum_{i=1}^{n_0} s_{iL}^{(0)}(x,h) \| F_{\widehat{\nu}_i^{(0)}}^{-1} - g \|_2^2 + \lambda \| g - \widehat{f}_{h,EM}(x) \|_2.$$

4: Projection to Wasserstein space

$$\widehat{m}_{L,h,EM}^{(0)}(x) = \arg\min_{\mu \in \mathcal{W}} \left\| F_{\mu}^{-1} - \widehat{f}_{0h,EM}(x) \right\|_2.$$

---

*where $\psi = \max_{1 \leq k \leq K} \| f^{(0)}(x) - f^{(k)}(x) \|_2$ quantifies the maximum discrepancy between the target and source.*

*Proof.* First note the following result has been established in [43]. $\| \widehat{f}_{EM}^{(k)}(x) - f^{(k)}(x) \|_2 = O_p(n_k^{-1/2} + N_{\min}^{-1/4})$.

Also, an analogy of Theorem 1 can be established in a similar way, just note that analysis in proof of 3 we have established that

$$\frac{1}{n} \sum_{k=1}^{K} \sum_{i=1}^{n_k} |s_{iG}^{(k)}(x)| = O_p(1).$$

Then using Theorem 7.9 in [3], we have

$$E\Big[\frac{1}{n}\sum_{k=1}^{K}\sum_{i=1}^{n_k}|s_{iG}^{(k)}(x)|\|F_{\nu_i^{(k)}}^{-1} - F_{\widehat{\nu}_i^{(k)}}^{-1}\|_2 \mid \{(x_i^{(0)},$$

$$\{y_{ij}^{(0)}\}_{j=1}^{N_i^{(0)}})\}_{i=1}^{n_0} \cup \big(\cup_{1\leq k\leq K}\{(x_i^{(k)}, \{y_{ij}^{(k)}\}_{j=1}^{N_i^{(k)}})\}_{i=1}^{n_k}\big)\Big]$$

$$\leq \frac{1}{2N_{\min}^{1/4}}\frac{1}{n}\sum_{k=1}^{K}\sum_{i=1}^{n_k}|s_{iG}^{(k)}(x)| = O_p(N_{\min}^{-1/4}).$$

Hence we have $\|\hat{f} - \hat{f}_{EM}\|_2 = O_p(N_{\min}^{-1/4})$. Then the augments in the proof of Theorem 2 still hold if $F_{\nu_i^{(0)}}^{-1}$, $\hat{f}^{(0)}$, $\hat{f}$ are replaced by their empirical-measure version. And the rate turns to be

$$O_p\Big((n_0^{-1/2} + N_{\min}^{-1/4})^{1-\epsilon}(\psi + \frac{\sum_{k=0}^{K}\sqrt{n_k}}{n_0+n_{\mathcal{A}}} + (n_0+n_{\mathcal{A}})^{-1/2} + N_{\min}^{-1/4})\Big). \qquad \square$$

**Theorem 7.** *Assume Condition 5 holds and the regularization parameter satisfies* $\lambda \asymp (n_0^{-1/2}h^{-1/2} + N_{\min}^{-1/4})^{1-\epsilon}$ *for some* $\epsilon > 0$. *Then, for the LWaTL-EM algorithm and a fixed* $x \in \mathbb{R}^p$, *it holds that*

$$d_{\mathcal{W}}^2(\widehat{m}_{L,h}^{(0)}(x), m^{(0)}(x))$$

$$= O_p\Big((n_0^{-1/2}h^{-1/2} + N_{\min}^{-1/4})^{1-\epsilon}(N_{\min}^{-1/4}$$

$$+ \psi_L + h^2 + h^{-1/2}\big(\sqrt{\sum_{k=0}^{K}\frac{1}{n_k}} + \sum_{k=0}^{K}\frac{\sqrt{n_k}}{n_0+n_{\mathcal{A}}}\big)\big]\Big),$$

*where* $\psi_L = \max_{1\leq k\leq K}\|f_{\oplus}^{(0)}(x) - f_{\oplus}^{(k)}(x)\|$.

*Proof.* Using the same arguments in the proof of Theorem 2 in [43] we can show that

$$\|f_h^{(0)}(x) - \widehat{f}_{h,EM}^{(0)}(x)\|_2 = O_p(n_0^{-1/2}h^{-1/2} + N_{\min}^{-1/4}).$$

Analysis in the proof of Theorem 4 shows

$$\frac{1}{n}\sum_{k=1}^{K}\sum_{i=1}^{n_k}|s_{iL}^{(k)}(x,h)| = O_p(1).$$

Then using Theorem 7.9 in [3], we have

$$E\Big[\frac{1}{n}\sum_{k=1}^{K}\sum_{i=1}^{n_k}|s_{iL}^{(k)}(x,h)|\|F_{\nu_i^{(k)}}^{-1} - F_{\widehat{\nu}_i^{(k)}}^{-1}\|_2 \mid \{(x_i^{(0)},$$

$$\{y_{ij}^{(0)}\}_{j=1}^{N_i^{(0)}})\}_{i=1}^{n_0} \cup \big(\cup_{1\leq k\leq K}\{(x_i^{(k)}, \{y_{ij}^{(k)}\}_{j=1}^{N_i^{(k)}})\}_{i=1}^{n_k}\big)\Big]$$

$$\leq \frac{1}{2N_{\min}^{1/4}}\frac{1}{n}\sum_{k=1}^{K}\sum_{i=1}^{n_k}|s_{iL}^{(k)}(x,h)| = O_p(N_{\min}^{-1/4}).$$

Hence we have $\|\hat{f}_h - \hat{f}_{h,EM}\|_2 = O_p(N_{\min}^{-1/4})$. Then the augments in the proof of Theorem 4 still hold if $F_{\nu_i^{(0)}}^{-1}$, $\hat{f}_h^{(0)}$, $\hat{f}_h$ are replaced by their empirical-measure version. And the rate turns to be

$$O_p\Big((n_0^{-1/2}h^{-1/2} + N_{\min}^{-1/4})^{1-\epsilon}\big[N_{\min}^{-1/4} + \psi_L + h^2 + h^{-1/2}\big(\sqrt{\sum_{k=0}^{K}\frac{1}{n_k}} + \sum_{k=0}^{K}\frac{\sqrt{n_k}}{n_0+n_{\mathcal{A}}}\big)\big]\Big). \qquad \square$$

