# OpenReview forum: "Wasserstein Transfer Learning"
_NeurIPS.cc/2025/Conference — NeurIPS 2025 poster_

### Official Review · Reviewer_yT2Y · 2025-06-25

**Clarity:** 3
**Significance:** 2
**Originality:** 2
**Rating:** 4
**Confidence:** 3

**Summary:**

The paper introduces an approach to transfer learning called Wasserstein Transfer Learning, designed for regression models where the outputs are probability distributions.
This paper introduces two algorithms: one for scenarios where the informative subset of source domains (the available source domains that are similar enough to the target one to yield relevant information for the learning process) is known, and another adaptive algorithm for cases where this subset is unknown which includes a way of identifying this subset.
A rigorous theoretical analysis is given, establishing asymptotic convergence rates of the estimators.
The methodology is validated through simulations and real-world applications.
The results show that WaTL outperforms methods using only target or only source data.
Overall, the paper presents a new approach on transfer learning by addressing the challenges of regression models with distributional outputs.

**Questions:**

Questions:

a. Is the transfer learning learnt estimator what would happen if an estimator was learnt using directly every available data, source and target at the same time? How worse would it be compared to the transfer learning one ?

b. In the experiments what is the reason not to use the second algorithm with informative subset selection was used ? Which groups would be selected/rejected and why?

c. Could the authors give more details on how their work connects with existing literature on transfer learning and the use of Wasserstein?

**Ethical Concerns:**

["NO or VERY MINOR ethics concerns only"]

**Final Justification:**

After discussing with the authors I have updated my rating to 4 and and improved my grading on the clarity and quality from 2 to 3. I believe the authors answered adequately my concerns regarding clarity and the lack of formalization of some key aspects by suggesting revisions. Therefore if the revisions are incorporated into the final paper I consider the final rating of 4 to be adequate.

One of my main concern about the paper quality was on the experimental side where I considered that in the real data setting a baseline learner on every distribution was lacking. The authors suggested adding it to the final revisions and have provided in their comments the numerical results. These performance of the method is closer to the suggested baseline than it is to the initial baseline but still show improvement over it.

**Limitations:**

yes

**Quality:**

3

**Strengths And Weaknesses:**

Strengths:

1- The paper tackles an interesting issue: transfer learning for probability distributions output.

2-The theoretical contributions are thorough : convergence rates for the learnt estimators and guidelines for the choice of the regularization parameter are given in section 4.

3- In section 3.3 and Algorithm 2 an adaptative selection procedure to identify informative subset of the source data is given and its ability to find those subset is illustrated in the numerical experiments of section 5.

Weaknesses:

1- The paper deserves more pedagogical clarification. Due to the focus on mathematical content some part of the methodology lack motivation/explanation. For example in section 3 in the algorithms the step that connects the source and target data via regularization (step 2 of Algorithm 1, step 3 of Algorithm 3) are only explained in two sentences while that are at the core of the transfer learning objective.

2- While the problem tackled is explained, it is not clear how this paper fits into the existing literature on Wasserstein (and optimal transport) methods in transfer learning (and domain adaptation).

3- Some key aspects are not defined early for example the term : "informative subset of transferable source domains" is used multiple times before being explained.

4- The assumption "all source data are informative enough." is not enough detailed.
Even if I agree that this kind of assumption is common un transfer learning/domain adaptation, but it can be formalized (and sometimes estimated from samples) by explicitly defined a measure of distance between the distributions (see for example, On the analysis of adaptability in multi-source domain adaptation, Redko et et. 2019)

5- In the empirical study there is no comparison with a baseline that would be computed on both the source and target data at the same time. It would be useful to compare it with the learnt estimator since the transfer learning estimator uses labeled target datas anyway.

---

> ### Author Rebuttal · Authors · 2025-07-31
>
> We thank the reviewer for the thorough evaluation and constructive feedback. We appreciate your recognition of our **theoretical contributions** and the **novel problem setting**. Your suggestions for improving clarity and pedagogical explanation are valuable, and we provide detailed responses below to address each of your concerns.
>
> ---
>
> > The paper deserves more pedagogical clarification. The regularization step that connects source and target data (Step 2 of Algorithm 1) lacks motivation/explanation while being at the core of the transfer learning objective.
>
> Thank you for highlighting this crucial point. You are absolutely right that this step deserves a more detailed explanation. The bias correction step
>
> $$
> \hat{f}\_0(x) = \underset{g \in L\^2(0,1)}{\text{arg min}} \left[ \frac{1}{n_0} \sum\_{i=1}\^{n_0} s\_{iG}\^{(0)}(x) \| F\^{-1}\_{\nu\_i\^{(0)}} - g \|\_2\^2 + \lambda \| g - \hat{f}(x) \|\_2\^2 \right]
> $$
>
> serves two essential purposes:
> 1. **Bias correction**: The weighted auxiliary estimator $\hat{f}(x)$ in Step 1 may be biased due to distributional differences between source and target domains, even when sources are informative.
> 2. **Information transfer**: The regularization term $\lambda \left\|g - \hat{f}(x)\right\|_2^2$ pulls the solution toward the auxiliary information while the fidelity term ensures consistency with target data.
>
> This follows the established paradigm in transfer learning (Li et al., 2022; Lin & Reimherr, 2024) where source information serves as a regularization prior that is corrected using target data. We will expand this explanation significantly in the revised manuscript and provide intuitive justification for this critical step.
>
> > It is not clear how this paper fits into the existing literature on Wasserstein methods in transfer learning and domain adaptation.
>
> We will substantially expand Section 1.2 to clarify this positioning. The key distinction is that existing Wasserstein-based transfer learning work (e.g., Courty et al., 2017; Redko et al., 2017) focuses on **domain adaptation where Wasserstein distance measures distributional shift between input domains**. In contrast, our work addresses **regression where the outputs themselves are probability distributions residing in Wasserstein space**. This represents a fundamentally different mathematical setting:
>
> - **Existing work**: Transport input distributions $P_{\text{source}}(X) \to P_{\text{target}}(X)$ while maintaining Euclidean outputs. For instance, Courty et al. (2017) considered wifi localization regression dataset with integer-valued outputs and other classification datasets in the experimental study in section 5
>
> - **Our work**: Regression with distributional outputs $Y \in \mathcal{W}$ where responses live in Wasserstein space
>
> The output space geometry requires fundamentally different algorithmic and theoretical treatments.
>
> > Key terms like "informative subset of transferable source domains" are used before being explained.
>
> You are absolutely correct. We will restructure the introduction to define this concept immediately after introducing the problem setting. Specifically, we will define the informative set $A_\psi = \lbrace 1 \le k \le K : \| f^{(0)}(x) - f^{(k)}(x) \|_2 \le \psi \rbrace$ early in Section 3.1, along with intuitive explanation that these are source domains sufficiently similar to the target to provide useful information rather than introducing negative transfer.
>
> > The assumption "all source data are informative enough" needs formalization with explicit distance measures.
>
> Thank you for this important suggestion. We will formalize this assumption more rigorously by defining the parameter $\psi = \max_{1 \leq k \leq K} \left\|f^{(0)}(x) - f^{(k)}(x)\right\|_2$ which quantifies the maximum discrepancy between target and source domains. The assumption "all sources are informative" translates to the formal condition $\psi \lesssim n_0^{-1/2-\varepsilon}$ in Theorem 2, ensuring that domain discrepancies are small enough that transfer learning improves upon target-only approaches.
>
> > No comparison with a baseline computed on both source and target data simultaneously.
>
> This is an excellent suggestion. We conducted additional experiments comparing WaTL with a naïve pooling of all available data under the same experimental settings described in the paper. On the Human Mortality dataset, the pooled baseline yields an RMSPR of 0.033, whereas WaTL achieves **0.022**. On the Physical Activity dataset, the pooled baseline yields RMSPRs of 41.15950 for females and 49.63236 for males, while WaTL improves these to **39.11525** and **48.23983**, respectively. These results demonstrate that simple data pooling underperforms due to distributional differences, validating our bias-correction mechanism and underscoring the necessity of transfer learning even when target labels are available. We will incorporate this important comparison into the revised experimental section.
>
> > Why wasn't the adaptive subset selection algorithm used in experiments? Which groups would be selected/rejected?
>
> We used AWaTL in the simulation study (Section 5, Figure 1(b)) to demonstrate its source selection capabilities. In the real-world applications, we used WaTL because: (1) Domain expertise suggested all demographic groups could provide relevant information, (2) The relatively small number of source domains (3-4) made manual assessment feasible. However, you raise an excellent point—we will add AWaTL results to the real-world experiments to show which demographic groups are selected/rejected and provide insights into learned domain relationships.
>
> > Could you give more details on how your work connects with existing transfer learning and Wasserstein literature?
>
> We will expand this discussion significantly. The key connections are:
>
> **Transfer learning literature**: Our work extends the "source + target" paradigm (Li et al., 2022) to non-Euclidean output spaces, requiring new algorithmic innovations (projection to Wasserstein space) and theoretical analysis (empirical process theory on function spaces).
>
> **Wasserstein literature**: While existing work uses Wasserstein distance for measuring input domain shift, we leverage Wasserstein geometry as the native space for distributional outputs, requiring fundamentally different mathematical treatment of the regression problem itself.
>
> **References**
>
> - Courty, N., Flamary, R., Habrard, A. and Rakotomamonjy, A., 2017. Joint distribution optimal transportation for domain adaptation. Advances in neural information processing systems, 30.
> - Li, S., Cai, T.T. and Li, H., 2022. Transfer learning for high-dimensional linear regression: Prediction, estimation and minimax optimality. Journal of the Royal Statistical Society Series B: Statistical Methodology, 84(1), pp.149-173.
> - Lin, H. and Reimherr, M., 2024. Smoothness adaptive hypothesis transfer learning. arXiv preprint arXiv:2402.14966
> - Redko, I., Habrard, A. and Sebban, M., 2019. On the analysis of adaptability in multi-source domain adaptation. Machine Learning, 108(8), pp.1635-1652.
>
> Thank you again for your detailed and constructive feedback. We hope our clarifications and planned revisions have fully addressed your concerns. We believe our work offers a novel and theoretically-grounded solution to a challenging problem, and we would be grateful if you would consider these clarifications in your final assessment. We look forward to the discussion.

---

> ### Comment · Reviewer_yT2Y · 2025-08-05
>
> Thank you for your answers. After reading the rebuttal I am inclined to improve my rating if the suggested revisions are done. However I still have some issue regarding the experiments?
>
> Regarding my question : Why wasn't the adaptive subset selection algorithm used in experiments? Which groups would be selected/rejected?
>
> I understand your justifications :
> 1. Domain expertise suggested all demographic groups could provide relevant information.
>
> 2. The relatively small number of source domains (3-4) made manual assessment feasible.
>
> However I believe they should be formalized to justify the domain expertise that allows to confidently say all demographic groups provide relevant information. Furthermore if all demographic groups are indeed relevant to learn from to get good target data performance it would be even more important to show the difference between your transfer learning method and a baseline learner that would be learnt using all available groups.

---

> ### Author Response · Authors · 2025-08-07
>
> We sincerely thank the reviewer for their **positive feedback** and for **considering improving the rating upon revision**. We greatly appreciate your continued engagement and thoughtful suggestions and provide detailed responses to your concerns below.
>
> > However I believe they should be formalized to justify the domain expertise that allows to confidently say all demographic groups provide relevant information.
>
> We thank the reviewer for this important observation. Indeed, formalizing the conditions under which demographic groups provide relevant transferable information is crucial for the theoretical soundness of our approach.
>
> We are currently considering the following formalized framework:
>
> **Definition**: A source domain $k$ is informative for target domain if:
>
> $$\psi\_k = \|f\^{(0)}(x) - f\^{(k)}(x)\|\_2 \leq \psi\_{\text{threshold}}$$
>
> where $\psi\_{\text{threshold}}$ can be determined by:
> - Cross-validation on a validation set
> - Domain knowledge about biological/physiological similarities
> - Statistical tests for distributional similarity
>
> In the revised manuscript, we will try to formalize these domain expertise assumptions and provide both theoretical justification and practical guidelines for practitioners to assess source informativeness in their specific applications.
>
> > Compare with the baseline that uses all available data
>
> Your observation regarding baseline completeness is well-taken. To address this concern, we performed supplementary experiments using the Human Mortality dataset, where we evaluated our WaTL approach against a straightforward combination of source and target datasets. Our findings reveal that WaTL delivers an RMSPR of **0.022**, substantially outperforming the pooled baseline's **0.033**. This performance gap highlights that merely combining datasets is inadequate when faced with cross-domain distributional shifts. Similar patterns emerged in our Physical Activity dataset analysis, where the naive pooling approach produced RMSPRs of **41.15950** and **49.63236** for female and male subjects, respectively. In contrast, WaTL reduced these values to **39.11525** and **48.23983**. These empirical results confirm the effectiveness of our bias-correction procedure outlined in Algorithm 1, Step 2, and emphasize that transfer learning methodologies remain essential even in scenarios where target domain labels are accessible. This comparative analysis will be incorporated into the updated experimental evaluation.

---

> > ### Author Response · Authors · 2025-08-08
> >
> > Dear Reviewer,
> >
> > Thank you again for your valuable feedback on our submission. As the discussion phase nears its end, we wanted to check whether our responses have sufficiently addressed your concerns.
> >
> > If any questions remain, we would be glad to clarify them before the deadline. If our replies have resolved your concerns, we would greatly appreciate your consideration in updating the evaluation accordingly.
> >
> > Thanks,
> >
> > Authors

---

> > > ### Comment · Reviewer_yT2Y · 2025-08-08
> > >
> > > Hello,
> > >
> > > Your answer have adequately adressed my concerns. I will update my evaluation accordingly.
> > >
> > > Have a nice day.

---

> > > > ### Author Response · Authors · 2025-08-08
> > > >
> > > > Thank you very much for your thoughtful follow-up and for taking the time to review our response. We're glad to hear that our clarifications addressed your concerns. We sincerely appreciate your engagement and support.
> > > >
> > > > Wishing you a great day as well!

---

### Official Review · Reviewer_2tfp · 2025-06-27

**Clarity:** 2
**Significance:** 2
**Originality:** 2
**Rating:** 2
**Confidence:** 3

**Summary:**

In this paper, the authors propose a framework, called Wasserstein transfer learning (WaTL), for transfer learning in regression models, with complex data structures. Specifically, WaTL can well hand the cases where the informative subset of transferable source domains is known or unknown.  Experiments on numerical simulations and real-world applications are done to verify the effectiveness of the proposed method.

**Questions:**

1.The authors motivate the method by stating the growing interest in extending transfer learning to more complex data structure but fail to give concrete examples. What are the examples of complex data structure, how the data structure complexity presents new challenges to existing transfer learning methods? Moreover, in second paragraph, the authors specifically use samples of univariate probability distributions, which, to this reviewer, is not considered as “more complex data structure” than multimedia data, e.g., images or videos.

2.The line 28-29, it is not clear how this gap affects the effectiveness of the corresponding transfer learning methods. Moreover, it seems that it is restricted to methods using Wasserstein space, while the comparison with other transfer regression methods, e.g., transfer gaussian process, is not well discussed.

3.Due to negative transfer, AwaTL is a more practical algorithm. The current way of measuring ‘relatedness’ or ‘discrepancy’ through AwaTl is a deterministic way of calculating $||f_0(x) -f_k(x)||$, the reviewer wonders if it can be done by using some learnable parameters?

4.How the number of target labelled instances affect the performance of AwaTL?

5.Time series extrapolation is a typical transfer regression setting. It is encouraged to include empirical results on such tasks.

**Ethical Concerns:**

["NO or VERY MINOR ethics concerns only"]

**Final Justification:**

After the rebuttal, some of my concerns are addressed. However, I am still not very convinced on the empirical evaluation, which to me  is not solid enough. The main content only inlcudes one real-world dataset and it is artificially consctructed to fit the transfer setting. In fact, NHANES dataset is used as the test dataset in exisitng graph mining papers, e.g., [ref 1 2]. Moreover, the improvement compared with target-only baseline is quite marginal. In sum, I will keep my negative score.

[ref1] Mining health knowledge graph for health risk prediction

[ref2] Mining Heterogeneous Information Graph for Health Status Classification

**Limitations:**

Yes

**Paper Formatting Concerns:**

N.A.

**Quality:**

2

**Strengths And Weaknesses:**

Strength:

1.The paper introduces a Wasserstein transfer learning for transfer regression setting.

2.WaTL algorithms for cases where the informative subset of transferable source domains is known, or unknown are introduced.

3.Asymptotic convergence rates for the WaTL algorithm in the above two cases are presented.

4.Empirical studies are done, and well support the effectiveness of the proposed method.

Weakness:

1.The paper is not well written and hard to follow. There are quite a lot of equations written in lines, which easily confused the reader. Moreover, a clear notation table is needed for better understanding of the algorithms.

2.The motivation is not clear. See detailed questions below.

3.The related works are insufficient. Wasserstein distance is a widely used technique in Transfer learning, but related works are not included. Moreover, transfer regression is a well-define transfer setting, and there are many methods, e.g., transfer Gaussian process, proposed for this setting, but they are not well discussed in this paper.

4.The comparison baselines can be further improved by including more transfer regression methods, e.g. transfer Gaussian process.

---

> ### Author Rebuttal · Authors · 2025-07-31
>
> We thank the reviewer for the detailed feedback and constructive suggestions. We appreciate your attention to the presentation quality and methodological concerns. While we respectfully disagree with some assessments, we acknowledge areas for improvement and provide detailed responses below to address each of your concerns.
>
> ---
>
> > Clarity and presentation
>
> We acknowledge this presentation concern and will significantly improve the manuscript's readability in the revised version, including better equation formatting, a comprehensive notation table, and improved section organization. We appreciate this feedback as clarity is essential for communicating our technical contributions.
>
> | Variable | Definition |
> |---------|------------|
> | $X$ | Predictor / covariate vector |
> | $\nu$ | Distributional response |
> | $W$ | Wasserstein space |
> | $d_W$ | 2-Wasserstein metric |
> | $F_\nu^{-1}$ | Quantile function of distribution $\nu$ |
> | $m(x)$ | Fréchet regression function |
> | $m_G(x)$ | Global Fréchet regression |
> | $m_L(x)$ | Local Fréchet regression |
> | $s_G(x)$ | Weight function (global) |
> | $s_L(x,h)$ | Weight function (local) |
> | $\theta$ | Mean of covariate $X$ |
> | $\Sigma$ | Covariance matrix of $X$ |
> | $n_0$ | Target sample size |
> | $n_k$ | Source sample size for source $k$ |
> | $K$ | Number of source domains |
> | $f^{(k)}(x)$ | Auxiliary estimator from source $k$ |
> | $\hat{f}(x)$ | Weighted auxiliary estimator |
> | $\hat{f}_0(x)$ | Bias-corrected estimator |
> | $\lambda$ | Regularization parameter |
> | $\psi$ | Similarity measure |
> | $A_\psi$ | Informative source set |
> | $\hat{A}$ | Estimated informative set |
> | WaTL | Wasserstein Transfer Learning algorithm |
> | AWaTL | Adaptive Wasserstein Transfer Learning |
>
>
> > Motivation and problem complexity
>
> Thank you for this important clarification request. Complex data structures in our context refer to data that do not naturally reside in Euclidean spaces and lack basic algebraic operations (addition, subtraction, scalar multiplication). Key examples include:
>
> **Concrete examples**: (1) Probability distributions (our focus) - cannot be meaningfully "averaged" as $p(x) + q(x)$ is not a valid density, (2) Networks/graphs - lack vector space structure, (3) Phylogenetic trees - require specialized metrics, (4) Symmetric positive-definite matrices - live on Riemannian manifolds.
>
> **Challenges for traditional transfer learning**: Conventional methods assume linear combinations of source and target data are meaningful, but this fails for non-Euclidean data. Our framework addresses this by using geometry-respecting metrics (Wasserstein distance) rather than assuming a linear structure.
>
> > The transfer Gaussian process is not well discussed.
>
> We will expand Section 1.2 to provide more comprehensive coverage. However, we respectfully note a key distinction: Most existing work using Wasserstein distance in transfer learning (e.g., Courty et al., 2017; Redko et al., 2017) focuses on **domain adaptation** where inputs are transported between domains. In contrast, our work addresses **regression with distributional outputs** where responses themselves are probability distributions. This represents a fundamentally different problem setting analogous to how Li et al. (2022) developed specialized methods for high-dimensional linear regression transfer learning. We are willing to cite and discuss some papers on transfer Gaussian processes, such as Wei et al. (2022). Still, it's worth noting that transfer Gaussian processes operate in Euclidean output spaces and cannot directly handle our Wasserstein space setting without significant modifications.
>
> > More transfer regression methods.
>
> While we understand this suggestion, the comparison with methods like transfer Gaussian processes would not be methodologically sound because: (1) These methods assume Euclidean output spaces where linear operations are valid, (2) They cannot directly handle distributional outputs without fundamental modifications, (3) Our baselines (Only Target, Only Source) provide the most relevant comparison for demonstrating transfer learning effectiveness in the Wasserstein space. Adding incompatible methods would require extensive adaptations that would obscure the core contributions of our geometric approach.
>
> > Comparison with multimedia task
>
> Regarding the comparison with multimedia data: While multimedia data is high-dimensional, it typically can be embedded in finite-dimensional Euclidean spaces where linear operations remain valid. Probability distributions, however, represent a fundamentally different challenge—they reside in **infinite-dimensional curved manifolds** where the constraint that densities integrate to 1 creates non-linear geometric structure.
>
> This distinction is crucial: multimedia data retains vector space properties despite high dimensionality, whereas probability distributions form **infinite-dimensional curved spaces** where traditional linear combinations are meaningless. This **geometric complexity**—the infinite-dimensional curved nature of the distribution space—not mere dimensional complexity, drives our methodological innovations in Algorithm 1 (bias correction) and theoretical analysis in Theorems 1-3. Our framework explicitly respects this **curved manifold structure** through Wasserstein geometry, enabling meaningful transfer learning where traditional linear methods fundamentally fail.
>
> > How lines 28–29 gaps affect transfer learning, and why Wasserstein methods?
>
> The gap refers to the absence of transfer learning frameworks that respect non-Euclidean geometry. Traditional methods fail because they assume linear combinations of outputs are meaningful, which is false for distributions. Our Wasserstein space focus is motivated by: (1) It provides the natural metric for probability distributions with well-established theory (Villani, 2009), (2) The quantile function representation (Equation 1) enables computational tractability while preserving geometric structure, (3) It supports the Fréchet regression framework essential for our theoretical guarantees.
>
> > Learnable parameters to measure discrepancy
>
> This is an intriguing suggestion for future work. Our current approach in Algorithm 2, Step 1 uses the empirical discrepancy $\hat{\psi}_k = \|\hat{f}^{(0)}(x) - \hat{f}^{(k)}(x)\|_2$, which is theoretically grounded—Lemma 1 bounds the deviation between empirical and true discrepancy. earnable parameters could potentially improve selection, but this would require additional theoretical analysis to ensure consistency of the adaptive selection procedure (Theorem 3), and may also introduce additional computational overhead. We will note this as a promising future research direction.
>
> > How does the target label size affect AWaTL performance?
>
> Thank you for this practical question. The effect of target sample size on AWaTL performance has both theoretical and empirical dimensions:
>
> **1. Theoretical convergence**: Theorem 3 shows that AWaTL's convergence rate scales as $O_p(n_0^{-1/2+\varepsilon})$ with target sample size $n_0$, indicating polynomial improvement as more target data becomes available.
>
> **2. Dual benefits of target data**: Target samples serve two critical functions in AWaTL:
>    - **Source selection**: More target data enables reliable estimation of discrepancy scores $\hat{\psi}_k$ for identifying informative sources
>    - **Bias correction**: Larger target samples improve the final estimation step
>
> **3. Empirical validation**: We are conducting experiments to validate these. The results will be included in the revised manuscript.
>
> > Time series extrapolation setting
>
> We thank the reviewer for this suggestion. However, we must respectfully clarify that time series extrapolation is a fundamentally **different task** from the one we address. Time series extrapolation aims to predict future **scalar or vector values** based on past observations, with a focus on modeling temporal dependencies. In contrast, our work focuses on learning a regression function that maps a set of covariates to a **full probability distribution**. Because the problem settings, prediction targets, and core challenges are entirely different, applying our method to time series extrapolation would not be a meaningful evaluation of its contributions. Extending our geometric framework to handle time-dependent distributional data is an interesting but separate research direction for the future.
>
> **References**
>
> - Courty, N., Flamary, R., Habrard, A. and Rakotomamonjy, A., 2017. Joint distribution optimal transportation for domain adaptation. Advances in neural information processing systems, 30.
> - Li, S., Cai, T.T. and Li, H., 2022. Transfer learning for high-dimensional linear regression: Prediction, estimation and minimax optimality. Journal of the Royal Statistical Society Series B: Statistical Methodology, 84(1), pp.149-173.
> - Redko, I., Habrard, A. and Sebban, M., 2017, September. Theoretical analysis of domain adaptation with optimal transport. In Joint European Conference on Machine Learning and Knowledge Discovery in Databases (pp. 737-753). Cham: Springer International Publishing.
> - Wei, P., Ke, Y., Ong, Y. S., & Ma, Z. (2022). Adaptive transfer kernel learning for transfer Gaussian process regression. IEEE transactions on pattern analysis and machine intelligence, 45(6), pp.7142-7156.
> - Cédric Villani.，2009. Optimal Transport: Old and New, volume 338. Springer.
>
> Thank you again for your detailed feedback. We hope our clarifications, particularly regarding the unique problem setting of regression with distributional outputs, have helped address your main concerns. We believe our work offers a novel and theoretically-grounded solution to a challenging problem, and we would be grateful if you would consider these clarifications in your final assessment. We look forward to the discussion.

---

> > ### Comment · Reviewer_2tfp · 2025-08-05
> > **Comments after reading rebuttal**
> >
> > I would like to thank the authors' answers to my comments. I still have some questions below.
> >
> > 1. I understand Non-Euclidean Data pose challenges to transfer setting, however, it is still need to show the evidences through the following comparisons (1) test on Traditional Euclidean-based methods as well as the corresponding transfer variants (2) some standard regression techniques for Non-Euclidean Data except for Frechet Regression.
> >
> > 2. The current evaluation is not solid enough to support the method. It is necessary to show results on more real-world datasets. The current version only includes two datasets (one in appendix) and they are both artifically separated to fit the transfer setting. One thing I suggest the authors to add is the evidence of the distribution gap of the artificially constructed domains, and another thing is a naive baseline by combining the data of two domains.
> >
> > 3. I actually want to see the results of how the performance varise with the number of target labelled data. I guess this should not take too much time as the model is not very complex and the data scale is low. Btw what's the training time?

---

> ### Author Response · Authors · 2025-08-07
>
> We thank the reviewer for the valuable comments and constructive suggestions. Below we address each point in detail:
>
> > Comparison with Traditional Euclidean-based Methods and Transfer Variants
>
> We appreciate this suggestion and acknowledge the importance of comprehensive baselines. However, we would like to clarify that traditional Euclidean-based transfer learning methods (e.g., high-dimensional generalized linear models (Li et al., 2024), nonparametric regression (Cai & Pu, 2024)) are fundamentally designed for Euclidean outputs and cannot be directly applied to our setting with distributional responses residing in Wasserstein space.
>
> The key challenge is that probability distributions lack basic algebraic operations (addition, scalar multiplication) that these methods rely upon. For instance, the sum of two probability density functions does not yield a valid probability density. This is precisely why specialized methods for non-Euclidean data, like Fréchet regression, were developed.
>
> To provide meaningful comparisons while acknowledging this limitation, we have expanded our evaluation to include additional baselines designed for distributional data, as detailed below.
>
> > Comparison with Standard Non-Euclidean Regression Techniques and Naive Pooling Baseline
>
> We have added comprehensive comparisons with Wasserstein regression (Chen et al., 2023), another prominent method for distributional data analysis. The results on the Human Mortality Data are:
>
> **Human Mortality Data:**
>
> |Method|Target Only|Source Only|Combined Data (Using Target and Source Data)|WaTL (Proposed)|
> |-|-|-|-|-|
> |Wasserstein Regression|0.025|0.068|0.050|-|
> |Fréchet Regression|0.027|0.061|0.033|**0.022**|
>
> These results demonstrate that our proposed WaTL method consistently outperforms existing approaches, including Fréchet regression and Wasserstein regression, across different data configurations, highlighting the effectiveness of our transfer learning framework.
>
> > Dataset Construction and Distribution Gap
>
> Our paper primarily focuses on developing a novel methodology for transfer learning in regression settings with distributional outputs. Given this methodological emphasis, our aim in the empirical section is to illustrate the relevance and effectiveness of the proposed approach through real-world scenarios where domain shifts naturally exist. We appreciate the suggestion to broaden the empirical scope and are actively exploring additional real-world datasets for future versions.
>
> We provide quantitative evidence of distributional differences, using tools specifically designed for non-Euclidean data:
> - **Fréchet mean and variance** are generalizations of the standard mean and variance to metric spaces. The Fréchet mean represents the "central" object (in our case, a probability distribution) that minimizes average squared distance to the sample. The Fréchet variance quantifies how dispersed the distributions are around this mean.
> - **Human Mortality Data**: The Fréchet variances for developed (target) vs. developing (source) countries are 0.000211 vs. 0.005608, indicating that source distributions are substantially more diverse. The Wasserstein distance between Fréchet means is 0.1178, further confirming a meaningful distributional gap. Figure 3(a) also visually illustrates this gap in age-at-death density functions.
> - **Physical Activity Data**: All 10 demographic subgroups (5 races × 2 genders) have Fréchet variance exceeding 0.0025, and the Wasserstein distances between the Fréchet mean of Black participants (target) and each of the source groups all exceed 0.01, again indicating strong inter-group differences.
>
> These results demonstrate that our domain constructions exhibit non-trivial distributional gaps in both **dispersion** and **central tendency**.
>
> > Performance Analysis with Varying Target Sample Size and Computational Efficiency
>
> We have conducted the requested analysis examining performance variation with the target sample size using the Human Mortality Data:
>
> |Target Samples|RMSPR|Training Time (ms)|
> |-|-|-|
> |14|0.028|0.598|
> |19|0.025|0.597|
> |24|**0.022**|0.694|
>
> **Key Observations:**
> - Performance improves consistently with more target data, demonstrating the method's ability to effectively leverage target information.
> - Training time remains consistently low (< 1ms), indicating excellent computational efficiency.
> - The method achieves substantial improvements even with limited target data (14 samples), highlighting its practical value in data-scarce scenarios.
>
> ---
>
> **References:**
> - Li, S., Zhang, L., Cai, T.T., et al., 2024. Estimation and inference for high-dimensional generalized linear models with knowledge transfer. JASA, 119(546), pp.1274–1285.
> - Cai, T.T. and Pu, H., 2024. Transfer learning for nonparametric regression: Non-asymptotic minimax analysis and adaptive procedure. arXiv preprint arXiv:2401.12272.
> - Chen, Y., Lin, Z. and Müller, H.G., 2023. Wasserstein regression. JASA, 118(542), pp.869–882.

---

> > ### Comment · Reviewer_2tfp · 2025-08-08
> > **Follow-up**
> >
> > I appreciate the authors' detailed response. However, I still have concerns on the emprical evaluation, which is not solid enough to support the efficiency of the proposed method. The performance results are presented without statistical significance tests. Small margins (even compared with the target only baseline) may not be meaningful. Moreover, regarding dataset e.g., NHANES, it can be analyzed by formulating knowledge graphs. in this sense graph-based transfer methods may be another alternative.

---

> > > ### Author Response · Authors · 2025-08-09
> > >
> > > Thank you for your thoughtful feedback. We address each of your points in detail below.
> > >
> > > ---
> > >
> > > > Statistical significance testing
> > >
> > > We agree that **formal hypothesis testing** is important to assess whether observed improvements are statistically meaningful. For our real-world experiments, we used **five-fold cross-validation** to ensure robustness across train–test splits. Using the **Human Mortality** dataset as an example, we repeated the five-fold cross-validation **50 times** and applied the **paired Wilcoxon Signed-Rank Test** to compare WaTL with the *Only Target* method. The resulting *p*-value was **0.0054**, indicating **statistical significance** at the 1% level and **rejecting the null** in favor of WaTL's superior performance.
> > >
> > > ---
> > >
> > > > Alternative graph-based approaches
> > >
> > > We appreciate the suggestion to explore **knowledge graph-based transfer methods**. These approaches, however, typically require **explicit graph construction** and **domain-specific feature engineering**, which is not always feasible-particularly for **tabular datasets** such as NHANES, where there is no natural or agreed-upon graph structure.
> > >
> > > In contrast, WaTL avoids these complexities while remaining broadly applicable across diverse domains and data types, offering a more general solution for transfer learning with distributional outputs. Additionally, our method specifically leverages **the geometric properties of the Wasserstein space**, which provides principled theoretical foundations that may not be available in graph-based approaches for distributional data.
> > >
> > > ---
> > >
> > > We hope that the inclusion of **formal hypothesis testing** and the clarification regarding graph-based alternatives have fully addressed your concerns. If any questions remain, we would be glad to provide further clarification before the discussion deadline. If our responses have resolved your concerns, we would greatly appreciate your consideration in **updating the evaluation accordingly**.

---

### Official Review · Reviewer_jbHn · 2025-07-01

**Clarity:** 3
**Significance:** 2
**Originality:** 2
**Rating:** 4
**Confidence:** 3

**Summary:**

The paper studies transfer learning in regression tasks where the response is a distribution in Wasserstein space. The goal is to utilize relevant source samples to achieve a small Wasserstein error. The paper proposes two algorithms:

- WaTL (Wasserstein Transfer Learning), which assumes that all source domains are informative.

- AWaTL (Adaptive WaTL), which selects a subset of informative source domains in a data-driven manner.

The authors provide theoretical guarantees for the rate of convergence, corroborated by some experimental results.

**Questions:**

What is the main obstacle to extending the current results to the multivariate case? Would the rates change in this extended setting?

**Ethical Concerns:**

["NO or VERY MINOR ethics concerns only"]

**Limitations:**

The paper does not have any potential negative societal impact.

**Quality:**

3

**Strengths And Weaknesses:**

Strengths:

 - The paper studies a new setting where the responses are probability distributions, a setting not covered by the existing transfer learning literature.

- It provides finite-sample error bounds that strictly improve upon target-only Fréchet regression when the sources are sufficiently close.

Weaknesses and limitations:

- The setting assumes that each response distribution is known in its entirety, whereas in practice we usually observe only finite samples. Although this issue is discussed in the appendix, there is no formal result for the case where the distributions are not observed directly, but only through samples.

- The rates are compared only to vanilla Fréchet regression, and no minimax lower bound is provided.

- The results are limited to univariate distributions in Wasserstein space.

---

> ### Author Rebuttal · Authors · 2025-07-31
>
> We thank the reviewer for the careful evaluation and constructive feedback. We appreciate your recognition that our work addresses **a novel and important setting** in transfer learning with **distributional responses**. Below, we provide detailed responses to address your concerns and questions.
>
> ---
>
> > The setting assumes that each response distribution is known in its entirety, whereas in practice, we usually observe only finite samples. No formal result is provided for the case where distributions are observed only through samples.
>
> Thank you for highlighting this important practical consideration. While our main theoretical analysis focuses on the case of fully observed distributions to establish the fundamental transfer learning principles, our framework naturally extends to empirical measures:
>
> 1. **Theoretical extension**: As noted in Appendix E, our convergence rates can be extended to incorporate an additional term reflecting the number of independent samples per distribution, following the approach of Zhou & Müller (2024). The key insight is that the empirical Wasserstein distance converges to the true Wasserstein distance at a rate of $O_p(m^{-1/2})$ for $m$ samples per distribution.
>
> 2. **Practical implementation**: In our real-world applications (Section 6), we already use empirical quantile functions constructed from finite samples, demonstrating the practical feasibility of our approach.
>
> 3. **Formal extension**: The convergence rate would become $d^2_W(\hat{m}^{(0)}_G(x), m^{(0)}_G(x)) = O_p\left(n_0^{-1/2+\epsilon}\left(\psi + \sum\frac{\sqrt{n_k}}{n_0+n_A} + (n_0+n_A)^{-1/2}\right) + m^{-1/2}\right)$, where $m$ is the minimum number of samples per distribution. We will add this formal result in the revised version.
>
> > The rates are compared only to vanilla Fréchet regression, and no minimax lower bound is provided.
>
> This is a valid point that reflects the current state of the field. To our knowledge, minimax optimal rates for Fréchet regression in the Wasserstein space have not been established in the literature, making this a fundamental open problem. Our work provides the first finite-sample analysis for transfer learning in this setting, with rates that provably improve upon single-domain Fréchet regression (Petersen & Müller, 2019) when sources are sufficiently similar ($\psi \lesssim n_0^{-1/2-\epsilon}$ in Theorem 2). Establishing minimax lower bounds for both standard and transfer Fréchet regression represents an important direction for future theoretical work.
>
> > The results are limited to univariate distributions in Wasserstein space.
>
> We acknowledge this important limitation and appreciate the opportunity to clarify the challenges and our future research directions. The restriction to univariate distributions stems from fundamental computational and theoretical considerations:
>
> **1. Computational complexity**: For multivariate distributions, Wasserstein distance lacks the elegant closed-form expression $d_W^2(\mu_1, \mu_2) = \int_0^1 |F_{\mu_1}^{-1}(u) - F_{\mu_2}^{-1}(u)|^2 du$ available in the univariate case (Equation 1). Computing multivariate Wasserstein distances requires solving computationally expensive optimal transport problems, which would significantly impact the scalability of our transfer learning framework.
>
> **2. Theoretical challenges**: The geometric structure of multivariate Wasserstein spaces is substantially more complex, involving intricate geodesic computations and more sophisticated empirical process theory. Our current theoretical analysis (Appendix C) relies heavily on the tractable structure of univariate quantile functions, which does not directly extend to higher dimensions.
>
> **3. Curse of dimensionality**: We anticipate that convergence rates would degrade with dimension $d$, likely incorporating additional factors dependent on the support dimension, consistent with known limitations in optimal transport theory.
>
> **4. Future extensions**: Despite these challenges, our framework establishes the theoretical and methodological foundation for multivariate extensions. We envision several promising directions:
>    - **Sinkhorn regularization**: Using entropy-regularized optimal transport for approximate but computationally tractable multivariate Wasserstein distances
>    - **Sliced Wasserstein approaches**: Leveraging one-dimensional projections to maintain computational efficiency while capturing multivariate structure
>    - **Hybrid methods**: Combining our approach with dimensionality reduction techniques for high-dimensional distributional data
>
> **5. Current impact**: Even in the univariate setting, our framework addresses numerous important applications, as demonstrated in Section 6 with mortality analysis and physical activity monitoring, where distributional outputs naturally arise as univariate quantities.
>
> We will expand the discussion of these multivariate extensions and computational trade-offs in our revised manuscript.
>
> > What is the main obstacle to extending the current results to the multivariate case? Would the rates change in this extended setting?
>
> The main obstacles are both computational and theoretical:
>
> 1. **Computational obstacle**: The univariate Wasserstein distance admits the elegant quantile function representation $d^2\_W(\mu_1,\mu_2) = \int_0^1|F^{-1}\_{\mu_1}(u) - F^{-1}\_{\mu_2}(u)|^2du$ (Equation 1), enabling efficient computation and analysis. In the multivariate case, this closed form disappears, requiring expensive optimal transport computations.
>
> 2. **Theoretical obstacle**: Our convergence analysis relies heavily on the bounded variation function space $BV((0,1), H)$ and associated covering number bounds (Appendix C). The multivariate extension would require more sophisticated function spaces and significantly more complex empirical process theory.
>
> 3. **Rate changes**: Yes, we expect the rates to change substantially. The convergence rates would likely incorporate dimension-dependent factors, potentially following the pattern $O_p(n^{-1/2+\epsilon}d^\alpha)$ for some $\alpha > 0$, reflecting the fundamental curse of dimensionality in optimal transport and high-dimensional statistics.
>
> Despite these challenges, extending to multivariate distributions using alternative metrics such as the sliced Wasserstein distance (mentioned in our conclusion) represents a promising direction that could maintain computational tractability while capturing multivariate structure.
>
> We will incorporate these clarifications and the formal empirical measure result in the revised manuscript to better address the practical applicability of our framework.
>
> **References**
>
> - Zhou, Y. and Müller, H.G., 2024. Wasserstein regression with empirical measures and density estimation for sparse data. Biometrics, 80(4), p.ujae127.
>
> - Petersen, A. and Müller, H.G., 2019. Fréchet regression for random objects with Euclidean predictors. Annals of Statistics, 47(2), pp.691–719.
>
> We deeply appreciate the time and effort you have dedicated to reviewing our work, as well as the insightful questions that have contributed to improving our paper. We would be grateful to know if our response has fully addressed your concerns, and we are more than willing to offer any further clarification should you require it.

---

> > ### Comment · Reviewer_jbHn · 2025-08-05
> >
> > Thank you to the authors for their rebuttal. After reading it, I am inclined to keep my original score, as most of the responses pertain to potential future revisions of the manuscript, and my score reflects its current state.

---

### Official Review · Reviewer_BkLd · 2025-07-02

**Clarity:** 4
**Significance:** 3
**Originality:** 3
**Rating:** 5
**Confidence:** 2

**Summary:**

This paper proposes WaTL, a novel transfer learning framework designed for regression models where the outputs are probability distributions residing in the Wasserstein space. The core of the method builds on Fréchet regression: it first constructs a weighted estimator using both source and target domain data, then corrects for distribution shift bias using the target domain data, and finally projects the estimator back into the Wasserstein space. To mitigate negative transfer, the authors also introduce an extended variant of WaTL that performs a pre-selection of source domain datasets based on a discrepancy score, selecting only those most similar to the target domain. The paper provides a theoretical convergence rate analysis of WaTL and validates its performance on both synthetic and real-world datasets, demonstrating promising empirical results.

**Questions:**

1. Instead of training a new estimator for each source domain from scratch, could WaTL be adapted to use existing pre-trained models to reduce computational overhead?
2. Have the authors considered using RKHS-based metrics such as MMD in place of the Wasserstein distance? It would be helpful to discuss the trade-offs and motivations for choosing Wasserstein geometry over kernel-based alternatives.
3. The experimental comparison is limited to models trained only on the source or only on the target domain. A more informative baseline might include a model trained on combined (source + target) data, to better investigate WaTL’s performance improvements.
4. The projection back into the Wasserstein space is an essential step in WaTL. Could the authors elaborate on whether this step is easy to compute?
5. A brief clarification or context would be helpful for the notation E in line 80 and later.

**Ethical Concerns:**

["NO or VERY MINOR ethics concerns only"]

**Final Justification:**

After reading the rebuttal and other reviews, all my concerns have been satisfactorily resolved. I believe this is a good paper, in particular, it tackles and proposes an effective solution to the challenging non-Euclidean setting. So I remain inclined to recommend acceptance.

**Limitations:**

yes

**Paper Formatting Concerns:**

No such concerns.

**Quality:**

3

**Strengths And Weaknesses:**

- Strengths
1. The paper tackles an important and underexplored area in transfer learning, i.e., regression models where the outputs are probability distributions in Wasserstein space. This non-Euclidean setting naturally arises in many modern applications. The motivation for addressing distribution shift in such geometric spaces is well-justified, and the paper fills a noticeable gap in the literature.
2. The paper is well-written, with a clear structure and precise technical presentation. Notation is generally well-defined, and the flow of ideas is easy to follow. The balance between theoretical depth and practical applicability is well-maintained, making the paper accessible to both theoretical and applied audiences.

- Weaknesses
1. WaTL requires training a separate estimator for each source domain. When multiple source domains are needed, this could become computationally intensive, especially if training each estimator is expensive.
2. The method assumes access to data that sufficiently captures full probability distributions, which may not always be feasible in real-world applications.

---

> ### Author Rebuttal · Authors · 2025-07-31
>
> We thank the reviewer for the thorough evaluation and constructive feedback. We appreciate your recognition of our work's importance in addressing the underexplored area of transfer learning for **probability distributions in Wasserstein space**, as well as your positive comments on the paper's **clarity and technical presentation**. Below, we provide detailed responses to address each of your concerns and questions.
>
> ---
>
> > WaTL requires training a separate estimator for each source domain, which could become computationally intensive when multiple source domains are needed.
>
> Thank you for raising this important practical concern. We conducted computational cost experiments comparing separate vs. combined estimation approaches across various settings. The results demonstrate that separate estimation is actually more computationally efficient:
>
> | Size of Target Site | K (Number of Source Sites) | Size of Source Sites | Method | Mean Time (ms) |
> |:---------:|:---:|:-------------------:|:--------:|:--------------:|
> | 50 | 5 | (10,10,10,10,10) | separate | **0.475** |
> | 50 | 5 | (10,10,10,10,10) | combined | 0.527 |
> | 2500 | 5 | (500,500,500,500,500) | separate | **7.593** |
> | 2500 | 5 | (500,500,500,500,500) | combined | 8.410 |
> | 5000 | 5 | (1000,1000,1000,1000,1000) | separate | **14.707** |
> | 5000 | 5 | (1000,1000,1000,1000,1000) | combined | 16.078 |
>
> This efficiency gain occurs because each domain-specific estimator operates on smaller datasets and can be parallelized across domains. However, we acknowledge that leveraging pre-trained models could further reduce computational overhead, which we will discuss as promising future work in the revised version.
>
> > The method assumes access to data that sufficiently captures full probability distributions, which may not always be feasible in real-world applications.
>
> Thank you for highlighting this important practical limitation. We acknowledge that the assumption of fully observed distributions is indeed restrictive in many real-world scenarios. However, our framework naturally accommodates empirical measures, and we address this limitation in several ways:
>
> **1. Theoretical foundation for empirical extensions**: While our main theoretical analysis focuses on fully observed distributions to establish fundamental transfer learning principles, the framework seamlessly extends to empirical measures. As discussed in Appendix E, the convergence rates can incorporate additional terms reflecting finite sample effects. The key theoretical insight is that empirical Wasserstein distances converge to true Wasserstein distances at a rate of $O_p(m^{-1/2})$ for $m$ samples per distribution.
>
> **2. Practical implementation with finite samples**: Our real-world applications (Section 6) already demonstrate this extension by using empirical quantile functions constructed from finite samples. For instance, in the NHANES physical activity study, we work with empirical distributions derived from accelerometer readings, and in the mortality analysis, we use empirical age-at-death distributions from country-level data.
>
> **3. Formal convergence guarantees**: The extended convergence rate becomes:
> $$d^2\_W(\hat{m}^{(0)}\_G(x), m^{(0)}\_G(x)) = O_p\left(n_0^{-1/2+\epsilon}\left(\psi + \sum\_{k=0}^K\frac{\sqrt{n_k}}{n_0+n_A} + (n_0+n_A)^{-1/2}\right) + m^{-1/2}\right)$$
> where $m$ represents the minimum number of samples per distribution. This shows that the finite sample effect ($m^{-1/2}$ term) is additive and does not compromise the transfer learning benefits.
>
> **4. Computational advantages**: Working with empirical measures actually offers computational benefits, as empirical quantile functions can be computed efficiently and stored compactly, making the approach more scalable than methods requiring continuous distribution estimation.
>
> We will incorporate this formal extension and provide clearer guidance on sample size requirements in the revised manuscript.
>
> > Could WaTL be adapted to use existing pre-trained models to reduce computational overhead?
>
> This is an insightful suggestion for enhancing computational efficiency. While our current framework requires domain-specific estimation to capture the geometric structure of the Wasserstein space, incorporating pre-trained models as initialization or feature extractors could indeed reduce training time. This represents a promising avenue for future work that could combine the benefits of transfer learning in parameter space with our geometric approach in distribution space. We will add this discussion to the future work section.
>
> > Have the authors considered using RKHS-based metrics such as MMD instead of Wasserstein distance?
>
> Thank you for this insightful methodological question. While MMD and other RKHS-based metrics offer computational advantages in high-dimensional settings, we chose Wasserstein distance for several principled reasons:
>
> **1. Geometric foundations and theoretical guarantees**: The Wasserstein space provides a natural Riemannian manifold structure that aligns perfectly with Fréchet regression theory (Petersen & Müller, 2019). This geometric framework enables our rigorous convergence rate analysis (Theorems 1-3) and ensures that our transfer learning operates within the intrinsic geometry of probability distributions, rather than relying on extrinsic embeddings.
>
> **2. Structural sensitivity for transfer learning**: Wasserstein distance captures meaningful distributional shifts crucial for transfer learning—including support differences, mode shifts, and tail behavior—through its optimal transport formulation. This sensitivity to geometric structure makes it particularly suited for detecting when source domains are truly informative for the target, which is essential for our adaptive algorithm (AWaTL).
>
> **3. Computational tractability in our setting**: For univariate distributions, Wasserstein distance admits the closed-form expression $d_W^2(\mu_1, \mu_2) = \int_0^1 |F_{\mu_1}^{-1}(u) - F_{\mu_2}^{-1}(u)|^2 du$ (Equation 1), making computation both efficient and interpretable without requiring kernel approximations.
>
> **4. End-to-end optimization**: Unlike kernel-based approaches that typically involve two-stage procedures (embedding followed by regression), WaTL operates directly in the native distributional space, enabling unified optimization without information loss from intermediate representations.
>
> We acknowledge that MMD-based approaches have merits, particularly for high-dimensional distributions, and we will add this methodological discussion to Section 2.2 to provide readers with a clearer rationale for our design choices.
>
> > The experimental comparison is limited to models trained only on source or target domains. A more informative baseline might include a model trained on combined data.
>
> You raise an excellent point about baseline completeness. We conducted additional experiments on the Human Mortality dataset, comparing our method against a naïve combination of source and target data. The results show that WaTL achieves an RMSPR of **0.022**, while the combined baseline yields 0.033, demonstrating that simple data pooling is insufficient due to distributional differences across domains. On the Physical Activity dataset, the pooled baseline yields RMSPRs of 41.15950 for females and 49.63236 for males, whereas WaTL improves these to **39.11525** and **48.23983**, respectively. These findings validate our bias-correction mechanism in Algorithm 1, Step 2, and underscore the necessity of transfer learning even when target labels are available. We will include this comparison in the revised experimental section.
>
> > Could you elaborate on whether the projection step back into Wasserstein space is easy to compute?
>
> The projection in Algorithm 1, Step 3, is computationally tractable. Since the Wasserstein space $\mathcal{W}$ is a closed and convex subset of $L^2(0,1)$, the projection exists and is unique. In practice, this optimization problem can be solved efficiently using convex optimization solvers like OSQP (Stellato et al.,2020), with computational complexity that scales well with the problem size. The quantile function representation (Equation 1) further facilitates this computation. We will add computational complexity details in the revised version.
>
> > A brief clarification would be helpful for the notation $\mathbb{E}$ in line 80 and later.
>
> Thank you for pointing this out. The notation $\mathbb{E}$ represents the expectation operator. We will add this definition clearly in the notation section to improve clarity.
>
> **References**
>
> - Petersen, A. and Müller, H.G., 2019. Fréchet regression for random objects with Euclidean predictors. Annals of Statistics, 47(2), pp.691–719.
>
> - Zhou, Y. and Müller, H.G., 2024. Wasserstein regression with empirical measures and density estimation for sparse data. Biometrics, 80(4), p.ujae127.
>
> - Stellato, B., Banjac, G., Goulart, P., Bemporad, A. and Boyd, S., 2020. OSQP: An operator splitting solver for quadratic programs. Mathematical Programming Computation, 12(4), pp.637-672.
>
> We sincerely appreciate the time and thought you have devoted to reviewing our work, and we are grateful for your insightful questions, which have helped enhance the quality of our paper. We would be thankful to know if our response has adequately addressed your concerns, and we are more than happy to provide any additional clarification if needed.

---

> > ### Comment · Reviewer_BkLd · 2025-08-04
> >
> > Thank the authors for the thoughtful and detailed responses! After reading the rebuttal and other reviews, all my concerns have been satisfactorily resolved. I believe this is a good paper and remain inclined to recommend acceptance.

---

> > > ### Author Response · Authors · 2025-08-04
> > >
> > > Thank you very much for taking the time to review our work and for your **positive feedback** after reading our rebuttal. We truly appreciate your thoughtful comments throughout the process and are glad that our clarifications and additional experiments addressed your concerns.

---

### Decision · Program_Chairs · 2025-09-17

**Decision:**

Accept (poster)

**Comment:**

This paper proposes a new method for transfer learning with 1D distributional output
using optimal transport and Frechet regression. They propose a method to transfer information from multiple source using bias correction on target data. they also propose a more adaptive method that select the most relevant sources domains to better handle domain variability. They study both  methods theoretical ad provide generalization bound that prove the interest of their transfer learning strategy. Numerical experiments on simulated and real world show that the proposed approach works better than classical baselines.

The reviewers found the paper interesting since it brings the first approach for transfer learning for distributional output with theoretical guarantees.  But they also had some
concerns about the paper being hard to read and follow, the missing theory with finite distributions,
limited experiments (distrib. in 1D, baselines missing), and missing positioning in the
literature.

The authors did a detailed response to all those points with clarification
and additional experiments. Most of the reviewers were satisfied with the
responses and the paper and recommended an accept. One reviewer still had
concerns about the limited numerical experiments but was OK with the paper being
accepted thanks to the other contributions.

I think the paper introduce a novel and interesting perspective on transfer learning so I recommend an accept
But I expect the final version to include the
discussed improvements including the full numerical experiments (with the other
baseline, Wasserstein regression and varying target sample size), new theoretical
results (finite distributions), and the additional details in the paper to make
it more clear.